# REPLICABLE BANDITS

**Hossein Esfandiari**
Google Research
esfandiari@google.com

**Alkis Kalavasis**
National Technical University of Athens
kalavasisalkis@mail.ntua.gr

**Amin Karbasi**
Yale University, Google Research
amin.karbasi@yale.edu

**Andreas Krause**
ETH Zurich
krausea@ethz.ch

**Vahab Mirrokni**
Google Research
mirrokni@google.com

**Grigoris Velegkas**
Yale University
grigoris.velegkas@yale.edu

## ABSTRACT

In this paper, we introduce the notion of replicable policies in the context of stochastic bandits, one of the canonical problems in interactive learning. A policy in the bandit environment is called replicable if it pulls, with high probability, the *exact* same sequence of arms in two different and independent executions (i.e., under independent reward realizations). We show that not only do replicable policies exist, but also they achieve almost the same optimal (non-replicable) regret bounds in terms of the time horizon. More specifically, in the stochastic multi-armed bandits setting, we develop a policy with an optimal problem-dependent regret bound whose dependence on the replicability parameter is also optimal. Similarly, for stochastic linear bandits (with finitely and infinitely many arms) we develop replicable policies that achieve the best-known problem-independent regret bounds with an optimal dependency on the replicability parameter. Our results show that even though randomization is crucial for the exploration-exploitation trade-off, an optimal balance can still be achieved while pulling the exact same arms in two different rounds of executions.

## 1 INTRODUCTION

In order for scientific findings to be valid and reliable, the experimental process must be repeatable, and must provide coherent results and conclusions across these repetitions. In fact, lack of reproducibility has been a major issue in many scientific areas; a 2016 survey that appeared in Nature (Baker, 2016a) revealed that more than 70% of researchers failed in their attempt to reproduce another researcher's experiments. What is even more concerning is that over 50% of them failed to reproduce their own findings. Similar concerns have been raised by the machine learning community, e.g., the ICLR 2019 Reproducibility Challenge (Pineau et al., 2019) and NeurIPS 2019 Reproducibility Program (Pineau et al., 2021), due to the to the exponential increase in the number of publications and the reliability of the findings.

The aforementioned empirical evidence has recently led to theoretical studies and rigorous definitions of replicability. In particular, the works of Impagliazzo et al. (2022) and Ahn et al. (2022) considered replicability as an algorithmic property through the lens of (offline) learning and convex optimization, respectively. In a similar vein, in the current work, we introduce the notion of replicability in the context of interactive learning and decision making. In particular, we study replicable policy design for the fundamental setting of stochastic bandits.

A multi-armed bandit (MAB) is a one-player game that is played over $T$ rounds where there is a set of different arms/actions $\mathcal{A}$ of size $|\mathcal{A}| = K$ (in the more general case of linear bandits, we can consider even an infinite number of arms). In each round $t = 1, 2, \ldots, T$, the player pulls an arm $a_t \in \mathcal{A}$ and receives a corresponding reward $r_t$. In the stochastic setting, the rewards of each

arm are sampled in each round independently, from some fixed but unknown, distribution supported on $[0, 1]$. Crucially, each arm has a potentially different reward distribution, but the distribution of each arm is fixed over time. A bandit algorithm $\mathbb{A}$ at every round $t$ takes as input the sequence of arm-reward pairs that it has seen so far, i.e., $(a_1, r_1), \ldots, (a_{t-1}, r_{t-1})$, then uses (potentially) some internal randomness $\xi$ to pull an arm $a_t \in \mathcal{A}$ and, finally, observes the associated reward $r_t \sim \mathcal{D}_{a_t}$.

We propose the following natural notion of a replicable bandit algorithm, which is inspired by the definition of Impagliazzo et al. (2022). Intuitively, a bandit algorithm is replicable if two distinct executions of the algorithm, with internal randomness fixed between both runs, but with independent reward realizations, give the exact same sequence of played arms, with high probability. More formally, we have the following definition.

**Definition 1** (Replicable Bandit Algorithm). *Let $\rho \in [0, 1]$. We call a bandit algorithm $\mathbb{A}$ $\rho$-replicable in the stochastic setting if for any distribution $\mathcal{D}_{a_j}$ over $[0, 1]$ of the rewards of the $j$-th arm $a_j \in \mathcal{A}$, and for any two executions of $\mathbb{A}$, where the internal randomness $\xi$ is shared across the executions, it holds that*

$$\Pr_{\xi, \boldsymbol{r}^{(1)}, \boldsymbol{r}^{(2)}} \left[ \left( a_1^{(1)}, \ldots, a_T^{(1)} \right) = \left( a_1^{(2)}, \ldots, a_T^{(2)} \right) \right] \geq 1 - \rho \,.$$

*Here, $a_t^{(i)} = \mathbb{A}(a_1^{(i)}, r_1^{(i)}, \ldots, a_{t-1}^{(i)}, r_{t-1}^{(i)}; \xi)$ is the $t$-th action taken by the algorithm $\mathbb{A}$ in execution $i \in \{1, 2\}$.*

The reason why we allow for some fixed internal randomness is that the algorithm designer has control over it, e.g., they can use the same seed for their (pseudo)random generator between two executions. Clearly, naively designing a replicable bandit algorithm is not quite challenging. For instance, an algorithm that always pulls the same arm or an algorithm that plays the arms in a particular random sequence determined by the shared random seed $\xi$ are both replicable. The caveat is that the performance of these algorithms in terms of expected regret will be quite poor. In this work, we aim to design bandit algorithms which are replicable and enjoy small expected regret. In the stochastic setting, the (expected) regret after $T$ rounds is defined as

$$\mathbf{E}[R_T] = T \max_{a \in \mathcal{A}} \mu_a - \mathbf{E}\left[ \sum_{t=1}^{T} \mu_{a_t} \right] \,,$$

where $\mu_a = \mathbf{E}_{r \sim \mathcal{D}_a}[r]$ is the mean reward for arm $a \in \mathcal{A}$. In a similar manner, we can define the regret in the more general setting of linear bandits (see, Section 5) Hence, the overarching question in this work is the following:

*Is it possible to design replicable bandit algorithms with small expected regret?*

At a first glance, one might think that this is not possible, since it looks like replicability contradicts the exploratory behavior that a bandit algorithm should possess. However, our main results answer this question in the affirmative and can be summarized in Table 1.

| Summary of Results | | | |
|---|---|---|---|
| Setting | Algorithm | Regret | Theorem |
| Stochastic MAB | Algorithm 1 | $\widetilde{O}\left( \frac{K^2 \log^3(T) H_\Delta}{\rho^2} \right)$ | Theorem 3 |
| Stochastic MAB | Algorithm 2 | $\widetilde{O}\left( \frac{K^2 \log(T) H_\Delta}{\rho^2} \right)$ | Theorem 4 |
| Stochastic Linear Bandits | Algorithm 3 | $\widetilde{O}\left( \frac{K^2 \sqrt{dT}}{\rho^2} \right)$ | Theorem 6 |
| Stochastic Linear Bandits Infinite Action Space | Algorithm 4 | $\widetilde{O}\left( \frac{\text{poly}(d)\sqrt{T}}{\rho^2} \right)$ | Theorem 10 |

Table 1: Our results for replicable stochastic general multi-armed and linear bandits. In the expected regret column, $\widetilde{O}(\cdot)$ subsumes logarithmic factors. $H_\Delta$ is equal to $\sum_{j:\Delta_j > 0} 1/\Delta_j$, $\Delta_j$ is the difference between the mean of action $j$ and the optimal action, $K$ is the number of arms, $d$ is the ambient dimension in the linear bandit setting.

## 1.1 RELATED WORK

**Reproducibility/Replicability.** In this work, we introduce the notion of replicability in the context of interactive learning and, in particular, in the fundamental setting of stochastic bandits. Close to our work, the notion of a replicable algorithm in the context of learning was proposed by Impagliazzo et al. (2022), where it is shown how any statistical query algorithm can be made replicable with a moderate increase in its sample complexity. Using this result, they provide replicable algorithms for finding approximate heavy-hitters, medians, and the learning of half-spaces. Reproducibility has been also considered in the context of optimization by Ahn et al. (2022). We mention that in Ahn et al. (2022) the notion of a replicable algorithm is different from our work and that of Impagliazzo et al. (2022), in the sense that the outputs of two different executions of the algorithm do not need to be exactly the same. From a more application-oriented perspective, Shamir & Lin (2022) study irreproducibility in recommendation systems and propose the use of smooth activations (instead of ReLUs) to improve recommendation reproducibility. In general, the reproducibility crisis is reported in various scientific disciplines Ioannidis (2005); McNutt (2014); Baker (2016b); Goodman et al. (2016); Lucic et al. (2018); Henderson et al. (2018). For more details we refer to the report of the NeurIPS 2019 Reproducibility Program Pineau et al. (2021) and the ICLR 2019 Reproducibility Challenge Pineau et al. (2019).

**Bandit Algorithms.** Stochastic multi-armed bandits for the general setting without structure have been studied extensively Slivkins (2019); Lattimore & Szepesvári (2020); Bubeck et al. (2012b); Auer et al. (2002); Cesa-Bianchi & Fischer (1998); Kaufmann et al. (2012a); Audibert et al. (2010); Agrawal & Goyal (2012); Kaufmann et al. (2012b). In this setting, the optimum regret achievable is $O\left(\log(T)\sum_{i:\Delta_i>0}\Delta^{-1}\right)$; this is achieved, e.g., by the upper confidence bound (UCB) algorithm of Auer et al. (2002). The setting of $d$-dimensional linear stochastic bandits is also well-explored Dani et al. (2008); Abbasi-Yadkori et al. (2011) under the well-specified linear reward model, achieving (near) optimal problem-independent regret of $O(d\sqrt{T\log(T)})$ Lattimore & Szepesvári (2020). Note that the best-known lower bound is $\Omega(d\sqrt{T})$ Dani et al. (2008) and that the number of arms can, in principle, be unbounded. For a finite number of arms $K$, the best known upper bound is $O(\sqrt{dT\log(K)})$ Bubeck et al. (2012a). Our work focuses on the design of replicable bandit algorithms and we hence consider only stochastic environments. In general, there is also extensive work in adversarial bandits and we refer the interested reader to Lattimore & Szepesvári (2020).

**Batched Bandits.** While sequential bandit problems have been studied for almost a century, there is much interest in the batched setting too. In many settings, like medical trials, one has to take a lot of actions in parallel and observe their rewards later. The works of Auer & Ortner (2010) and Cesa-Bianchi et al. (2013) provided sequential bandit algorithms which can easily work in the batched setting. The works of Gao et al. (2019) and Esfandiari et al. (2021) are focusing exclusively on the batched setting. Our work on replicable bandits builds upon some of the techniques from these two lines of work.

## 2 STOCHASTIC BANDITS AND REPLICABILITY

In this section, we first highlight the main challenges in order to guarantee replicability and then discuss how the results of Impagliazzo et al. (2022) can be applied in our setting.

### 2.1 WARM-UP I: NAIVE REPLICABILITY AND CHALLENGES

Let us consider the stochastic two-arm setting ($K = 2$) and a bandit algorithm $\mathbb{A}$ with two independent executions, $\mathbb{A}_1$ and $\mathbb{A}_2$. The algorithm $\mathbb{A}_i$ plays the sequence $1, 2, 1, 2, \ldots$ until some, potentially random, round $T_i \in \mathbb{N}$ after which one of the two arms is eliminated and, from that point, the algorithm picks the winning arm $j_i \in \{1, 2\}$. The algorithm $\mathbb{A}$ is $\rho$-replicable if and only if $T_1 = T_2$ and $j_1 = j_2$ with probability $1 - \rho$.

Assume that $|\mu_1 - \mu_2| = \Delta$ where $\mu_i$ is the mean of the distribution of the $i$-th arm. If we assume that $\Delta$ is known, then we can run the algorithm for $T_1 = T_2 = \frac{C}{\Delta^2}\log(1/\rho)$ for some universal constant $C > 0$ and obtain that, with probability $1 - \rho$, it will hold that $\widehat{\mu}_1^{(j)} \approx \mu_1$ and $\widehat{\mu}_2^{(j)} \approx \mu_2$

for $j \in \{1, 2\}$, where $\widehat{\mu}_i^{(j)}$ is the estimation of arm's $i$ mean during execution $j$. Hence, knowing $\Delta$ implies that the stopping criterion of the algorithm $\mathbb{A}$ is deterministic and that, with high probability, the winning arm will be detected at time $T_1 = T_2$. This will make the algorithm $\rho$-replicable.

Observe that when $K = 2$, the only obstacle to replicability is that the algorithm should decide at the same time to select the winning arm and the selection must be the same in the two execution threads. In the presence of multiple arms, there exists the additional constraint that the above conditions must be satisfied during, potentially, multiple arm eliminations. Hence, the two questions arising from the above discussion are (i) how to modify the above approach when $\Delta$ is unknown and (ii) how to deal with $K > 2$ arms.

A potential solution to the second question (on handling $K > 2$ arms) is the Execute-Then-Commit (ETC) strategy. Consider the stochastic $K$-arm bandit setting. For any $\rho \in (0, 1)$, the ETC algorithm with known $\Delta = \min_i \Delta_i$ and horizon $T$ that uses $m = \frac{4}{\Delta^2} \log(1/\rho)$ deterministic exploration phases before commitment is $\rho$-replicable. The intuition is exactly the same as in the $K = 2$ case. The caveats of this approach are that it assumes that $\Delta$ is known and that the obtained regret is quite unsatisfying. In particular, it achieves regret bounded by $m \sum_{i \in [K]} \Delta_i + \rho \cdot (T - mK) \sum_{i \in [k]} \Delta_i$.

Next, we discuss how to improve the regret bound without knowing the gaps $\Delta_i$. Before designing new algorithms, we will inspect the guarantees that can be obtained by combining ideas from previous results in the bandits literature and the recent work in replicable learning of Impagliazzo et al. (2022).

## 2.2 WARM-UP II: BANDIT ALGORITHMS AND REPLICABLE MEAN ESTIMATION

First, we remark that we work in the stochastic setting and the distributions of the rewards of the two arms are subgaussian. Thus, the problem of estimating their mean is an instance of a statistical query for which we can use the algorithm of Impagliazzo et al. (2022) to get a replicable mean estimator for the distributions of the rewards of the arms.

**Proposition 2** (Replicable Mean Estimation (Impagliazzo et al., 2022)). *Let* $\tau, \delta, \rho \in [0, 1]$. *There exists a $\rho$-replicable algorithm* ReprMeanEstimation *that draws* $\Omega\left(\frac{\log(1/\delta)}{\tau^2(\rho-\delta)^2}\right)$ *samples from a distribution with mean $\mu$ and computes an estimate $\widehat{\mu}$ that satisfies $|\widehat{\mu} - \mu| \leq \tau$ with probability at least $1 - \delta$.*

Notice that we are working in the regime where $\delta \ll \rho$, so the sample complexity is $\Omega\left(\frac{\log(1/\delta)}{\tau^2 \rho^2}\right)$.

The straightforward approach is to try to use an optimal multi-armed algorithm for the stochastic setting, such as UCB or arm-elimination (Even-Dar et al., 2006), combined with the replicable mean estimator. However, it is not hard to see that this approach does not give meaningful results: if we want to achieve replicability $\rho$ we need to call the replicable mean estimator routine with parameter $\rho/(KT)$, due to the union bound that we need to take. This means that we need to pull every arm at least $K^2 T^2$ times, so the regret guarantee becomes vacuous. This gives us the first key insight to tackle the problem: we need to reduce the number of calls to the mean estimator. Hence, we will draw inspiration from the line of work in stochastic batched bandits (Gao et al., 2019; Esfandiari et al., 2021) to derive *replicable* bandit algorithms.

## 3 REPLICABLE MEAN ESTIMATION FOR BATCHED BANDITS

As a first step, we would like to show how one could combine the existing replicable algorithms of Impagliazzo et al. (2022) with the batched bandits approach of Esfandiari et al. (2021) to get some preliminary non-trivial results. We build an algorithm for the $K$-arm setting, where the gaps $\Delta_j$ are unknown to the learner. Let $\delta$ be the confidence parameter of the arm elimination algorithm and $\rho$ be the replicability guarantee we want to achieve. Our approach is the following: let us, deterministically, split the time interval into sub-intervals of increasing length. We treat each subinterval as a batch of samples where we pull each active arm the same number of times and use the replicable mean estimation algorithm to, empirically, compute the true mean. At the end of each batch, we decide to eliminate some arm $j$ using the standard UCB estimate. Crucially, if we condition on the event that all the calls to the replicable mean estimator return the same number, then the algorithm we propose is replicable.

---

**Algorithm 1** Mean-Estimation Based Replicable Algorithm for Stochastic MAB (Theorem 3)

---
1: **Input:** time horizon $T$, number of arms $K$, replicability $\rho$
2: **Initialization:** $B \leftarrow \log(T), q \leftarrow T^{1/B}, c_0 \leftarrow 0, \mathcal{A} \leftarrow [K], r \leftarrow T, \widehat{\mu}_a \leftarrow 0, \forall a \in \mathcal{A}$
3: **for** $i = 1$ **to** $B - 1$ **do**
4:      **if** $\lfloor q^i \rfloor \cdot |\mathcal{A}| > r$ **then**
5:          **break**
6:      $c_i = c_{i-1} + \lfloor q^i \rfloor$
7:      Pull every arm $a \in \mathcal{A}$ for $\lfloor q^i \rfloor$ times
8:      **for** $a \in \mathcal{A}$ **do**
9:          $\widehat{\mu}_a \leftarrow \texttt{ReprMeanEstimation}(\delta = 1/(2KTB), \tau = 1, \sqrt{\log(2KTB)/c_i}, \rho' = \rho/(KB))$          ▷ Proposition 2
10:      $r \leftarrow r - |\mathcal{A}| \cdot \lfloor q^i \rfloor$
11:      **for** $a \in \mathcal{A}$ **do**
12:          **if** $\widehat{\mu}_a < \max_{a \in \mathcal{A}} \widehat{\mu}_a - 2\tau$ **then**
13:              **Remove** $a$ from $\mathcal{A}$
14: In the last batch play the arm from $\mathcal{A}$ with the smallest index

---

**Theorem 3.** *Let $T \in \mathbb{N}, \rho \in (0, 1]$. There exists a $\rho$-replicable algorithm (presented in Algorithm 1) for the stochastic bandit problem with $K$ arms and gaps $(\Delta_j)_{j \in [K]}$ whose expected regret is*

$$\mathbf{E}[R_T] \leq C \cdot \frac{K^2 \log^2(T)}{\rho^2} \sum_{j: \Delta_j > 0} \left( \Delta_j + \frac{\log(KT \log(T))}{\Delta_j} \right),$$

*where $C > 0$ is an absolute numerical constant, and its running time is polynomial in $K, T$ and $1/\rho$.*

The above result, whose proof can be found in Appendix A, states that, by combining the tools from Impagliazzo et al. (2022) and Esfandiari et al. (2021), we can design a replicable bandit algorithm with (instance-dependent) expected regret $O(K^2 \log^3(T)/\rho^2)$. Notice that the regret guarantee has an extra $K^2 \log^2(T)/\rho^2$ factor compared to its non-replicable counterpart in Esfandiari et al. (2021) (Theorem 5.1). This is because, due to a union bound over the rounds and the arms, we need to call the replicable mean estimator with parameter $\rho/(K \log(T))$. In the next section, we show how to get rid of the $\log^2(T)$ by designing a new algorithm.

## 4 IMPROVED ALGORITHMS FOR REPLICABLE STOCHASTIC BANDITS

While the previous result provides a non-trivial regret bound, it is not optimal with respect to the time horizon $T$. In this section, we show how to improve it by designing a new algorithm, presented in Algorithm 2, which satisfies the guarantees of Theorem 4 and, essentially, decreases the dependence on the time horizon $T$ from $\log^3(T)$ to $\log(T)$. Our main result for replicable stochastic multi-armed bandits with $K$ arms follows.

**Theorem 4.** *Let $T \in \mathbb{N}, \rho \in (0, 1]$. There exists a $\rho$-replicable algorithm (presented in Algorithm 2) for the stochastic bandit problem with $K$ arms and gaps $(\Delta_j)_{j \in [K]}$ whose expected regret is*

$$\mathbf{E}[R_T] \leq C \cdot \frac{K^2}{\rho^2} \sum_{j: \Delta_j > 0} \left( \Delta_j + \frac{\log(KT \log(T))}{\Delta_j} \right),$$

*where $C > 0$ is an absolute numerical constant, and its running time is polynomial in $K, T$ and $1/\rho$.*

Note that, compared to the non-replicable setting, we incur an extra factor of $K^2/\rho^2$ in the regret. The proof can be found in Appendix B. Let us now describe how Algorithm 2 works. We decompose the time horizon into $B = \log(T)$ batches. Without the replicability constraint, one could draw $q^i$ samples in batch $i$ from each arm and estimate the mean reward. With the replicability constraint, we have to boost this: in each batch $i$, we pull each active arm $O(\beta q^i)$ times, for some $q$ to be determined, where $\beta = O(K^2/\rho^2)$ is the replicability blow-up. Using these samples, we compute

---

**Algorithm 2** Replicable Algorithm for Stochastic Multi-Armed Bandits (Theorem 4)

---
1: **Input:** time horizon $T$, number of arms $K$, replicability $\rho$
2: **Initialization:** $B \leftarrow \log(T)$, $q \leftarrow T^{1/B}$, $c_0 \leftarrow 0$, $\mathcal{A}_0 \leftarrow [K]$, $r \leftarrow T$, $\widehat{\mu}_a \leftarrow 0, \forall a \in \mathcal{A}_0$
3: $\beta \leftarrow \lfloor \max\{K^2/\rho^2, 2304\} \rfloor$
4: **for** $i = 1$ **to** $B - 1$ **do**
5:     **if** $\beta \lfloor q^i \rfloor \cdot |\mathcal{A}_i| > r$ **then**
6:         **break**
7:     $\mathcal{A}_i \leftarrow \mathcal{A}_{i-1}$
8:     **for** $a \in \mathcal{A}_i$ **do**
9:         Pull arm $a$ for $\beta \lfloor q^i \rfloor$ times
10:         Compute the empirical mean $\widehat{\mu}_\alpha^{(i)}$
11:     $c_i \leftarrow c_{i-1} + \lfloor q^i \rfloor$
12:     $\widetilde{c}_i \leftarrow \beta c_i$
13:     $\widetilde{U}_i \leftarrow \sqrt{2\ln(2KTB)/\widetilde{c}_i}$
14:     $U_i \leftarrow \sqrt{2\ln(2KTB)/c_i}$
15:     $\overline{U}_i \leftarrow \mathrm{Uni}[U_i/2, U_i]$
16:     $r \leftarrow r - \beta \cdot |\mathcal{A}_i| \cdot \lfloor q^i \rfloor$
17:     **for** $a \in \mathcal{A}_i$ **do**
18:         **if** $\widehat{\mu}_a^{(i)} + \widetilde{U}_i < \max_{a \in \mathcal{A}_i} \widehat{\mu}_a^{(i)} - \overline{U}_i$ **then**
19:             **Remove** $a$ from $\mathcal{A}_i$
20: In the last batch play the arm from $\mathcal{A}_{B-1}$ with the smallest index

---

the empirical mean $\widehat{\mu}_\alpha^{(i)}$ for any active arm $\alpha$. Note that $\widetilde{U}_i$ in Algorithm 2 corresponds to the size of the actual confidence interval of the estimation and $U_i$ corresponds to the confidence interval of an algorithm that does not use the $\beta$-blow-up in the number of samples. The novelty of our approach comes from the choice of the interval around the mean of the maximum arm: we pick a threshold uniformly at random from an interval of size $U_i/2$ around the maximum mean. Then, the algorithm checks whether $\widehat{\mu}_a^{(i)} + \widetilde{U}_i < \max \widehat{\mu}_{a'}^{(i)} - \overline{U}_i$, where $\max$ runs over the active arms $a'$ in batch $i$, and eliminates arms accordingly. To prove the result we show that there are three regions that some arm $j$ can be in relative to the confidence interval of the best arm in batch $i$ (cf. Appendix B). If it lies in two of these regions, then the decision of whether to keep it or discard it is the same in both executions of the algorithm. However, if it is in the third region, the decision could be different between parallel executions, and since it relies on some external and unknown randomness, it is not clear how to reason about it. To overcome this issue, we use the random threshold to argue about the probability that the decision between two executions differs. The crucial observation that allows us to get rid of the extra $\log^2(T)$ factor is that there are correlations between consecutive batches: we prove that if some arm $j$ lies in this "bad" region in some batch $i$, then it will be outside this region after a constant number of batches.

## 5 REPLICABLE STOCHASTIC LINEAR BANDITS

We now investigate replicability in the more general setting of stochastic linear bandits. In this setting, each arm is a vector $a \in \mathbb{R}^d$ belonging to some action set $\mathcal{A} \subseteq \mathbb{R}^d$, and there is a parameter $\theta^\star \in \mathbb{R}^d$ unknown to the player. In round $t$, the player chooses some action $a_t \in \mathcal{A}$ and receives a reward $r_t = \langle \theta^\star, a_t \rangle + \eta_t$, where $\eta_t$ is a zero-mean 1-subgaussian random variable independent of any other source of randomness. This means that $\mathbf{E}[\eta_t] = 0$ and satisfies $\mathbf{E}[\exp(\lambda \eta_t)] \leq \exp(\lambda^2/2)$ for any $\lambda \in \mathbb{R}$. For normalization purposes, it is standard to assume that $\|\theta^\star\|_2 \leq 1$ and $\sup_{a \in \mathcal{A}} \|a\|_2 \leq 1$. In the linear setting, the expected regret after $T$ pulls $a_1, \ldots, a_T$ can be written as

$$\mathbf{E}[R_T] = T \sup_{a \in \mathcal{A}} \langle \theta^\star, a \rangle - \mathbf{E}\left[\sum_{t=1}^{T} \langle \theta^\star, a_t \rangle\right].$$

In Section 5.1 we provide results for the finite action space case, i.e., when $|\mathcal{A}| = K$. Next, in Section 5.2, we study replicable linear bandit algorithms when dealing with infinite action spaces. In the following, we work in the regime where $T \gg d$. We underline that our approach leverages connections of stochastic linear bandits with G-optimal experiment design, core sets constructions, and least-squares estimators. Roughly speaking, the goal of G-optimal design is to find a (small) subset of arms $\mathcal{A}'$, which is called the core set, and define a distribution $\pi$ over them with the following property: for any $\varepsilon > 0, \delta > 0$ pulling only these arms for an appropriate number of times and computing the least-squares estimate $\widehat{\theta}$ guarantees that $\sup_{a \in \mathcal{A}} \langle a, \theta^* - \widehat{\theta} \rangle \le \varepsilon$, with probability $1 - \delta$. For an extensive discussion, we refer to Chapters 21 and 22 of Lattimore & Szepesvári (2020).

## 5.1 FINITE ACTION SET

We first introduce a lemma that allows us to reduce the size of the action set that our algorithm has to search over.

**Lemma 5** (See Chapters 21 and 22 in Lattimore & Szepesvári (2020)). *For any finite action set $\mathcal{A}$ that spans $\mathbb{R}^d$ and any $\delta, \varepsilon > 0$, there exists an algorithm that, in time polynomial in $d$, computes a multi-set of $\Theta(d \log(1/\delta)/\varepsilon^2 + d \log \log d)$ actions (possibly with repetitions) such that (i) they span $\mathbb{R}^d$ and (ii) if we perform these actions in a batched stochastic $d$-dimensional linear bandits setting with true parameter $\theta^\star \in \mathbb{R}^d$ and let $\widehat{\theta}$ be the least-squares estimate for $\theta^\star$, then, for any $a \in \mathcal{A}$, with probability at least $1 - \delta$, we have $\left| \left\langle a, \theta^\star - \widehat{\theta} \right\rangle \right| \le \varepsilon$.*

Essentially, the multi-set in Lemma 5 is obtained using an approximate *G-optimal design* algorithm. Thus, it is crucial to check whether this can be done in a replicable manner. Recall that the above set of distinct actions is called the core set and is the solution of an (approximate) G-optimal design problem. To be more specific, consider a distribution $\pi : \mathcal{A} \to [0, 1]$ and define $V(\pi) = \sum_{a \in \mathcal{A}} \pi(a) a a^\top \in \mathbb{R}^{d \times d}$ and $g(\pi) = \sup_{a \in \mathcal{A}} \|a\|_{V(\pi)^{-1}}^2$. The distribution $\pi$ is called a design and the goal of G-optimal design is to find a design that minimizes $g$. Since the number of actions is finite, this problem reduces to an optimization problem which can be solved efficiently using standard optimization methods (e.g., the Frank-Wolfe method). Since the initialization is the same, the algorithm that finds the optimal (or an approximately optimal) design is replicable under the assumption that the gradients and the projections do not have numerical errors. This perspective is orthogonal to the work of Ahn et al. (2022), that defines reproducibility from a different viewpoint.

---

**Algorithm 3** Replicable Algorithm for Stochastic Linear Bandits (Theorem 6)

1: Input:  number of arms $K$, time horizon $T$, replicability $\rho$
2: Initialization: $B \leftarrow \log(T), q \leftarrow (T/c)^{1/B}, \mathcal{A} \leftarrow [K], r \leftarrow T$
3: $\beta \leftarrow \lfloor \max\{K^2/\rho^2, 2304\} \rfloor$
4: **for** $i = 1$ **to** $B - 1$ **do**
5:     $\widetilde{\varepsilon}_i = \sqrt{d \log(KT^2)/(\beta q^i)}$
6:     $\varepsilon_i = \sqrt{d \log(KT^2)/q^i}$
7:     $n_i = 10 d \log(KT^2)/\varepsilon_i^2$
8:     $a_1, \ldots, a_{n_i} \leftarrow$ multi-set given by Lemma 5 with parameters $\delta = 1/(KT^2)$ and $\varepsilon = \widetilde{\varepsilon}_i$
9:     **if** $n_i > r$ **then**
10:         **break**
11:     Pull every arm $a_1, \ldots, a_{n_i}$ and receive rewards $r_1, \ldots, r_{n_i}$
12:     Compute the LSE $\widehat{\theta}_i \leftarrow \left( \sum_{j=1}^{n_i} a_j a_j^T \right)^{-1} \left( \sum_{j=1}^{n_i} a_j r_j \right)$
13:     $\bar{\varepsilon}_i \leftarrow \text{Uni}[\varepsilon_i/2, \varepsilon_i]$
14:     $r \leftarrow r - n_i$
15:     **for** $a \in \mathcal{A}$ **do**
16:         **if** $\langle a, \widehat{\theta}_i \rangle + \widetilde{\varepsilon}_i < \max_{a \in \mathcal{A}} \langle a, \widehat{\theta}_i \rangle - \bar{\varepsilon}_i$ **then**
17:             **Remove** $a$ from $\mathcal{A}$
18: In the last batch play $\arg \max_{a \in \mathcal{A}} \langle a, \widehat{\theta}_{B-1} \rangle$

---

In our batched bandit algorithm (Algorithm 3), the multi-set of arms $a_1, \ldots, a_{n_i}$ computed in each batch is obtained via a deterministic algorithm with runtime $\text{poly}(K, d)$, where $|\mathcal{A}| = K$. Hence, the

multi-set will be the same in two different executions of the algorithm. On the other hand, the LSE will not be since it depends on the stochastic rewards. We apply the techniques that we developed in the replicable stochastic MAB setting in order to design our algorithm. Our main result for replicable $d$-dimensional stochastic linear bandits with $K$ arms follows. For the proof, we refer to Appendix C.

**Theorem 6.** *Let $T \in \mathbb{N}, \rho \in (0,1]$. There exists a $\rho$-replicable algorithm for the stochastic $d$-dimensional linear bandit problem with $K$ arms whose expected regret is*

$$\mathbf{E}[R_T] \leq C \cdot \frac{K^2}{\rho^2} \sqrt{dT \log(KT)} \,,$$

*where $C > 0$ is an absolute numerical constant, and its running time is polynomial in $d, K, T$ and $1/\rho$.*

Note that the best known non-replicable algorithm achieves an upper bound of $\widetilde{O}(\sqrt{dT \log(K)})$ and, hence, our algorithm incurs a replicability overhead of order $K^2/\rho^2$. The intuition behind the proof is similar to the multi-armed bandit setting in Section 4.

## 5.2 INFINITE ACTION SET

Let us proceed to the setting where the action set $\mathcal{A}$ is unbounded. Unfortunately, even when $d = 1$, we cannot directly get an algorithm that has satisfactory regret guarantees by discretizing the space and using Algorithm 3. The approach of Esfandiari et al. (2021) is to discretize the action space and use an $1/T$-net to cover it, i.e. a set $\mathcal{A}' \subseteq A$ such that for all $a \in \mathcal{A}$ there exists some $a' \in \mathcal{A}'$ with $||a - a'||_2 \leq 1/T$. It is known that there exists such a net of size at most $(3T)^d$ (Vershynin, 2018, Corollary 4.2.13). Then, they apply the algorithm for the finite arms setting, increasing their regret guarantee by a factor of $\sqrt{d}$. However, our replicable algorithm for this setting contains an additional factor of $K^2$ in the regret bound. Thus, even when $d = 1$, our regret guarantee is greater than $T$, so the bound is vacuous. One way to fix this issue and get a sublinear regret guarantee is to use a smaller net. We use a $1/T^{1/(4d+2)}-$net that has size at most $(3T)^{\frac{d}{4d+2}}$ and this yields an expected regret of order $O(T^{4d+1/(4d+2)}\sqrt{d \log(T)}/\rho^2)$. For further details, we refer to Appendix D.

Even though the regret guarantee we managed to get using the smaller net of Appendix D is sublinear in $T$, it is not a satisfactory bound. The next step is to provide an algorithm for the infinite action setting using a replicable LSE subroutine combined with the batching approach of Esfandiari et al. (2021). We will make use of the next lemma.

**Lemma 7** (Section 21.2 Note 3 of Lattimore & Szepesvári (2020)). *There exists a deterministic algorithm that, given an action space $\mathcal{A} \subseteq \mathbb{R}^d$, computes a 2-approximate G-optimal design $\pi$ with a core set of size $O(d \log \log(d))$.*

We additionally prove the next useful lemma, which, essentially, states that we can assume without loss of generality that every arm in the support of $\pi$ has mass at least $\Omega(1/(d \log(d)))$. We refer to Appendix F.1 for the proof.

**Lemma 8** (Effective Support). *Let $\pi$ be the distribution that corresponds to the 2-approximate optimal G-design of Lemma 7 with input $\mathcal{A}$. Assume that $\pi(a) \leq c/(d \log(d))$, where $c > 0$ is some absolute numerical constant, for some arm $a$ in the core set. Then, we can construct a distribution $\widehat{\pi}$ such that, for any arm $a$ in the core set, $\widehat{\pi}(a) \geq C/(d \log(d))$, where $C > 0$ is an absolute constant, so that it holds*

$$\sup_{a' \in \mathcal{A}} \|a'\|_{V(\widehat{\pi})^{-1}}^2 \leq 4d \,.$$

The upcoming lemma is a replicable algorithm for the least-squares estimator and, essentially, builds upon Lemma 7 and Lemma 8. Its proof can be found at Appendix F.2.

**Lemma 9** (Replicable LSE). *Let $\rho, \varepsilon \in (0,1]$ and $0 < \delta \leq \min\{\rho, 1/d\}$[1]. Consider an environment of $d$-dimensional stochastic linear bandits with infinite action space $\mathcal{A}$. Assume that $\pi$ is a 4-approximate optimal design with associated core set $\mathcal{C}$ as computed by Lemma 7 with input $\mathcal{A}$. There exists a $\rho$-replicable algorithm that pulls each arm $a \in \mathcal{C}$ a total of*

$$\Omega \left( \frac{d^4 \log(d/\delta) \log^2 \log(d) \log \log \log(d)}{\varepsilon^2 \rho^2} \right)$$

---

[1]We can handle the case of $0 < \delta \leq d$ by paying an extra $\log d$ factor in the sample complexity.

*times and outputs $\theta_{\mathrm{SQ}}$ that satisfies $\sup_{a \in \mathcal{A}} |\langle a, \theta_{\mathrm{SQ}} - \theta^\star \rangle| \leq \varepsilon$, with probability at least $1 - \delta$.*

---

**Algorithm 4** Replicable LSE Algorithm for Stochastic Infinite Action Set (Theorem 10)

---

1: Input: time horizon $T$, action set $\mathcal{A} \subseteq \mathbb{R}^d$, replicability $\rho$
2: $\mathcal{A}' \leftarrow 1/T$-net of $\mathcal{A}$
3: Initialization: $r \leftarrow T, B \leftarrow \log(T), q \leftarrow (T/c)^{1/B}$
4: **for** $i = 1$ **to** $B - 1$ **do**
5: $\quad$ $q^i$ denotes the number of pulls of all arms before the replicability blow-up
6: $\quad$ $\varepsilon_i = c \cdot d\sqrt{\log(T)/q^i}$
7: $\quad$ The blow-up is $M_i = q^i \cdot d^3 \log(d) \log^2 \log(d) \log \log \log(d) \log^2(T)/\rho^2$
8: $\quad$ $a_1, \ldots, a_{|\mathcal{C}_i|} \leftarrow$ core set $\mathcal{C}_i$ of the design given by Lemma 7 with parameter $\mathcal{A}'$
9: $\quad$ **if** $\lceil M_i \rceil > r$ **then**
10: $\quad\quad$ **break**
11: $\quad$ Pull every arm $a_j$ for $N_i = \lceil M_i \rceil/|\mathcal{C}_i|$ rounds and receive rewards $r_1^{(j)}, \ldots, r_{N_i}^{(j)}$ for $j \in [|\mathcal{C}_i|]$
12: $\quad$ $S_i = \{(a_j, r_t^{(j)}) : t \in [N_i], j \in [|\mathcal{C}_i|]\}$
13: $\quad$ $\widehat{\theta}_i \leftarrow \texttt{ReplicableLSE}(S_i, \rho' = \rho/(dB), \delta = 1/(2|\mathcal{A}'|T^2), \tau = \min\{\varepsilon_i, 1\})$
14: $\quad$ $r \leftarrow r - \lceil M_i \rceil$
15: $\quad$ **for** $a \in \mathcal{A}'$ **do**
16: $\quad\quad$ **if** $\langle a, \widehat{\theta}_i \rangle < \max_{a \in \mathcal{A}'} \langle a, \widehat{\theta}_i \rangle - 2\varepsilon_i$ **then**
17: $\quad\quad\quad$ **Remove** $a$ from $\mathcal{A}'$
18: In the last batch play $\arg\max_{a \in \mathcal{A}'} \langle a, \widehat{\theta}_{B-1} \rangle$
19:
20: $\texttt{ReplicableLSE}(S, \rho, \delta, \tau)$
21: **for** $a \in \mathcal{C}$ **do**
22: $\quad$ $v(a) \leftarrow \texttt{ReplicableSQ}(\phi : x \in \mathbb{R} \mapsto x \in \mathbb{R}, S, \rho, \delta, \tau)$ $\qquad \triangleright$ Impagliazzo et al. (2022)
23: **return** $(\sum_{j \in |S|} a_j a_j^\top)^{-1} \cdot (\sum_{a \in \mathcal{C}} a \, n_a \, v(a))$

---

The main result for the infinite actions' case, obtained by Algorithm 4, follows. Its proof can be found at Appendix E.

**Theorem 10.** *Let $T \in \mathbb{N}, \rho \in (0, 1]$. There exists a $\rho$-replicable algorithm (Algorithm 4) for the stochastic $d$-dimensional linear bandit problem with infinite action set whose expected regret is*

$$\mathbf{E}[R_T] \leq C \cdot \frac{d^4 \log(d) \log^2 \log(d) \log \log \log(d)}{\rho^2} \sqrt{T} \log^{3/2}(T),$$

*where $C > 0$ is an absolute numerical constant, and its running time is polynomial in $T^d$ and $1/\rho$.*

Our algorithm for the infinite arm linear bandit case enjoys an expected regret of order $\widetilde{O}(\mathrm{poly}(d)\sqrt{T})$. We underline that the dependence of the regret on the time horizon is (almost) optimal, and we incur an extra $d^3$ factor in the regret guarantee compared to the non-replicable algorithm of Esfandiari et al. (2021). We now comment on the time complexity of our algorithm.

**Remark 11.** *The current implementation of our algorithm requires time exponential in $d$. However, for a general convex set $\mathcal{A}$, given access to a separation oracle for it and an oracle that computes an (approximate) G-optimal design, we can execute it in polynomial time and with polynomially many calls to the oracle. Notably, when $\mathcal{A}$ is a polytope such oracles exist. We underline that computational complexity issues also arise in the traditional setting of linear bandits with an infinite number of arms and the computational overhead that the replicability requirement adds is minimal. For further details, we refer to Appendix G.*

## 6 CONCLUSION AND FUTURE DIRECTIONS

In this paper, we have provided a formal notion of reproducibility/replicability for stochastic bandits and we have developed algorithms for the multi-armed bandit and the linear bandit settings that satisfy this notion and enjoy a small regret decay compared to their non-replicable counterparts. We hope and believe that our paper will inspire future works in replicable algorithms for more complicated interactive learning settings such as reinforcement learning. We also provide experimental evaluation in Appendix H.

## 7 ACKNOWLEDGEMENTS

Alkis Kalavasis was supported by the Hellenic Foundation for Research and Innovation (H.F.R.I.) under the "First Call for H.F.R.I. Research Projects to support Faculty members and Researchers and the procurement of high-cost research equipment grant", project BALSAM, HFRIFM17-1424. Amin Karbasi acknowledges funding in direct support of this work from NSF (IIS-1845032), ONR (N00014- 19-1-2406), and the AI Institute for Learning-Enabled Optimization at Scale (TILOS). Andreas Krause was supported by the European Research Council (ERC) under the European Union's Horizon 2020 research and innovation program grant agreement no. 815943 and the Swiss National Science Foundation under NCCR Automation, grant agreement 51NF40 180545. Grigoris Velegkas was supported by NSF (IIS-1845032), an Onassis Foundation PhD Fellowship and a Bodossaki Foundation PhD Fellowship.

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

# A  THE PROOF OF THEOREM 3

**Theorem.** *Let $T \in \mathbb{N}, \rho \in (0,1]$. There exists a $\rho$-replicable algorithm (presented in Algorithm 1) for the stochastic bandit problem with $K$ arms and gaps $(\Delta_j)_{j \in [K]}$ whose expected regret is*

$$\mathbf{E}[R_T] \leq C \cdot \frac{K^2 \log^2(T)}{\rho^2} \sum_{j:\Delta_j>0} \left( \Delta_j + \frac{\log(2KT\log(T))}{\Delta_j} \right),$$

*where $C > 0$ is an absolute numerical constant, and its running time is polynomial in $K, T$ and $1/\rho$.*

*Proof.* First, we claim that the algorithm is $\rho$-replicable: since the elimination decisions are taken in the same iterates and are based solely on the mean estimations, the replicability of the algorithm of Proposition 2 implies the replicability of the whole algorithm. In particular,

$$\mathbf{Pr}[(a_1,...,a_T) \neq (a'_1,...,a'_T)] = \mathbf{Pr}[\exists i \in [B], \exists j \in [K]: \widehat{\mu}_j^{(i)} \text{ was not replicable}] \leq \rho.$$

During each batch $i$, we draw for any active arm $\lfloor q^i \rfloor$ fresh samples for a total of $c_i$ samples and use the replicable mean estimation algorithm to estimate its mean. For an active arm, at the end of some batch $i \in [B]$, we say that its estimation is "correct" if the estimation of its mean is within $\sqrt{\log(2KTB)/c_i}$ from the true mean. Using Proposition 2, the estimation of any active arm at the end of any batch (except possibly the last batch) is correct with probability at least $1 - 1/(2KTB)$ and so, by the union bound, the probability that the estimation is incorrect for some arm at the end of some batch is bounded by $1/T$. We remark that when $\delta < \rho$, the sample complexity of Proposition 2 reduces to $O(\log(1/\delta)/(\tau^2\rho^2))$. Let $\mathcal{E}$ denote the event that our estimates are correct. The total expected regret can be bounded as

$$\mathbf{E}[R_T] \leq T \cdot 1/T + \mathbf{E}[R_T|\mathcal{E}].$$

It suffices to bound the second term of the RHS and hence we can assume that each gap is correctly estimated within an additive factor of $\sqrt{\log(2KTB)/c_i}$ after batch $i$. First, due to the elimination condition, we get that the best arm is never eliminated. Next, we have that

$$\mathbf{E}[R_T|\mathcal{E}] = \sum_{j:\Delta_j>0} \Delta_j \, \mathbf{E}[T_j|\mathcal{E}],$$

where $T_j$ is the total number of pulls of arm $j$. Fix a sub-optimal arm $j$ and assume that $i+1$ was the last batch it was active. Since this arm is not eliminated at the end of batch $i$, and the estimations are correct, we have that

$$\Delta_j \leq \sqrt{\log(2KTB)/c_i},$$

and so $c_i \leq \log(2KTB)/\Delta_j^2$. Hence, the number of pulls to get the desired bound due to Proposition 2 is (since we need to pull an arm $c_i/\rho_1^2$ times in order to get an estimate at distance $\sqrt{\log(1/\delta)/c_i^2}$ with probability $1 - \delta$ in a $\rho_1$-replicable manner when $\delta < \rho_1$)

$$T_j \leq c_{i+1}/\rho_1^2 = q/\rho_1^2(1+c_i) \leq q/\rho_1^2 \cdot (1 + \log(2KTB)/\Delta_j^2).$$

This implies that the total regret is bounded by

$$\mathbf{E}[R_T] \leq 1 + q/\rho_1^2 \cdot \sum_{j:\Delta_j>0} \left( \Delta_j + \frac{\log(2KTB)}{\Delta_j} \right).$$

We finally set $q = T^{1/B}$ and $B = \log(T)$. Moreover, we have that $\rho_1 = \rho/(KB)$. These yield

$$\mathbf{E}[R_T] \leq \frac{K^2 \log^2(T)}{\rho^2} \sum_{j:\Delta_j>0} \left( \Delta_j + \frac{\log(2KT\log(T))}{\Delta_j} \right).$$

This completes the proof. ◻

# B  THE PROOF OF THEOREM 4

**Theorem.** *Let $T \in \mathbb{N}, \rho \in (0,1]$. There exists a $\rho$-replicable algorithm (presented in Algorithm 2) for the stochastic bandit problem with $K$ arms and gaps $(\Delta_j)_{j \in [K]}$ whose expected regret is*

$$\mathbf{E}[R_T] \leq C \cdot \frac{K^2}{\rho^2} \sum_{j:\Delta_j > 0} (\Delta_j + \log(KT\log(T))/\Delta_j) \,,$$

*for some absolute numerical constant $C > 0$, and its running time is polynomial in $K, T$ and $1/\rho$.*

To give some intuition, we begin with a non tight analysis which, however, provides the main ideas behind the actual proof.

**Non Tight Analysis**   Assume that the environment has $K$ arms with unknown means $\mu_i$ and let $T$ be the number of rounds. Consider $B$ to the total number of batches and $\beta > 1$. We set $q = T^{1/B}$. In each batch $i \in [B]$, we pull each arm $\beta \lfloor q^i \rfloor$ times. Hence, after the $i$-th batch, we will have drawn $\widetilde{c}_i = \sum_{1 \leq j \leq i} \beta \lfloor q^j \rfloor$ independent and identically distributed samples from each arm. Let us also set $c_i = \sum_{1 \leq j \leq i} \lfloor q^j \rfloor$.

Let us fix $i \in [B]$. Using Hoeffding's bound for subgaussian concentration, the length of the confidence bound for arm $j \in [K]$ that guarantees $1 - \delta$ probability of success (in the sense that the empirical estimate $\widehat{\mu}_j$ will be close to the true $\mu_j$) is equal to

$$\widetilde{U}_i = \sqrt{2\log(1/\delta)/\widetilde{c}_i} \,,$$

when the estimator uses $\widetilde{c}_i$ samples. Also, let

$$U_i = \sqrt{2\log(1/\delta)/c_i} \,.$$

Assume that the active arms at the batch iteration $i$ lie in the set $\mathcal{A}_i$. Consider the estimates $\{\widehat{\mu}_j^{(i)}\}_{i \in [B], j \in \mathcal{A}_i}$, where $\widehat{\mu}_j^{(i)}$ is the empirical mean of arm $j$ using $\widetilde{c}_i$ samples. We will eliminate an arm $j$ at the end of the batch iteration $i$ if

$$\widehat{\mu}_j^{(i)} + \widetilde{U}_i \leq \max_{t \in \mathcal{A}_i} \widehat{\mu}_t^{(i)} - \overline{U}_i \,,$$

where $\overline{U}_i \sim \mathrm{Uni}[U_i/2, U_i]$. For the remaining of the proof, we condition on the event $\mathcal{E}$ that for every arm $j \in [K]$ and every batch $i \in [B]$ the true mean is within $\widetilde{U}_i$ from the empirical one.

We first argue about the replicability of our algorithm. Consider a fixed round $i$ (end of $i$-th batch) and a fixed arm $j$. Let $i^\star$ be the optimal empirical arm after the $i$-th batch.

Let $\widehat{\mu}_j^{(i)'}, \widehat{\mu}_{i^\star}^{(i)'}$ the empirical estimates of arms $j, i^\star$ after the $i$-th batch, under some other execution of the algorithm. We condition on the event $\mathcal{E}'$ for the other execution as well. Notice that $|\widehat{\mu}_j^{(i)'} - \widehat{\mu}_j^{(i)}| \leq 2\widetilde{U}_i, |\widehat{\mu}_{i^\star}^{(i)'} - \widehat{\mu}_{i^\star}^{(i)}| \leq 2\widetilde{U}_i$. Notice that, since the randomness of $\overline{U}_i$ is shared, if $\widehat{\mu}_j^{(i)} + \widetilde{U}_i \geq \widehat{\mu}_{i^\star}^{(i)} - \overline{U}_i + 4\widetilde{U}_i$, then the arm $j$ will not be eliminated after the $i$-th batch in some other execution of the algorithm as well. Similarly, if $\widehat{\mu}_j^{(i)} + \widetilde{U}_i < \widehat{\mu}_{i^\star}^{(i)} - \overline{U}_i - 4\widetilde{U}_i$ the the arm $j$ will get eliminated after the $i$-th batch in some other execution of the algorithm as well. In particular, this means that if $\widehat{\mu}_j^{(i)} - 2\widetilde{U}_i > \widehat{\mu}_{i^\star}^{(i)} + \widetilde{U}_i - U_i/2$ then the arm $j$ will not get eliminated in some other execution of the algorithm and if $\widehat{\mu}_j^{(i)} + 5\widetilde{U}_i < \widehat{\mu}_{i^\star}^{(i)} - U_i$ then the arm $j$ will also get eliminated in some other execution of the algorithm with probability 1 under the event $\mathcal{E} \cap \mathcal{E}'$. We call the above two cases good since they preserve replicability. Thus, it suffices to bound the probability that the decision about arm $j$ will be different between the two executions when we are in neither of these cases. Then, the worst case bound due to the mass of the uniform probability measure is

$$\frac{16\sqrt{2\log(1/\delta)/\widetilde{c}_i}}{\sqrt{2\log(1/\delta)/c_i}} \,.$$

This implies that the probability mass of the bad event is at most $16\sqrt{c_i/\widetilde{c}_i} = 16\sqrt{1/\beta}$. A union bound over all arms and batches yields that the probability that two distinct executions differ in at least one pull is

$$\mathbf{Pr}[(a_1, \ldots, a_T) \neq (a_1', \ldots, a_T')] \leq 16KB\sqrt{1/\beta} + 2\delta \,,$$

and since $\delta \leq \rho$ it suffices to pick $\beta = 768K^2B^2/\rho^2$.

We now focus on the regret of our algorithm. Let us set $\delta = 1/(KTB)$. Fix a sub-optimal arm $j$ and assume that batch $i+1$ was the last batch that is was active. We obtain that the total number of pulls of this arm is

$$T_j \leq \widetilde{c}_{i+1} \leq \beta q(1+c_i) \leq \beta q(1 + 8\log(1/\delta)/\Delta_j^2)$$

From the replicability analysis, it suffices to take $\beta$ of order $K^2\log^2(T)/\rho^2$ and so

$$\mathbf{E}[R_T] \leq T\cdot 1/T + \mathbf{E}[R_T|\mathcal{E}] = 1 + \sum_{j:\Delta_j>0} \Delta_j\,\mathbf{E}[T_j|\mathcal{E}] \leq \frac{C\cdot K^2\log^2(T)}{\rho^2}\sum_{j:\Delta_j>0}\left(\Delta_j + \frac{\log(KT\log(T))}{\Delta_j}\right),$$

for some absolute constant $C > 0$.

Notice that the above analysis, which uses a naive union bound, does not yield the desired regret bound. We next provide a more tight analysis of the same algorithm that achieves the regret bound of Theorem 4.

**Improved Analysis** (*The Proof of Theorem 4*) In the previous analysis, we used a union bound over all arms and all batches in order to control the probability of the bad event. However, we can obtain an improved regret bound as follows. Fix a sub-optimal arm $i \in [K]$ and let $t$ be the first round that it appears in the bad event. We claim that after a constant number of rounds, this arm will be eliminated. This will shave the $O(\log^2(T))$ factor from the regret bound. Essentially, as indicated in the previous proof, the bad event corresponds to the case where the randomness of the cut-off threshold $\overline{U}$ can influence the decision of whether the algorithm eliminates an arm or not. The intuition is that during the rounds $t$ and $t+1$, given that the two intervals intersected at round $t$, we know that the probability that they intersect again is quite small since the interval of the optimal mean is moving upwards, the interval of the sub-optimal mean is concentrating around the guess and the two estimations have been moved by at most a constant times the interval's length.

Since the bad event occurs at round $t$, we know that

$$\widehat{\mu}_j^{(t)} \in \left[\widehat{\mu}_{t^\star}^{(t)} - U_t - 5\widetilde{U}_t, \widehat{\mu}_{t^\star}^{(t)} - U_t/2 + 3\widetilde{U}_t\right].$$

In the above $\widehat{\mu}_{t^\star}^t$ is the estimate of the optimal mean at round $t$ whose index is denoted by $t^\star$. Now assume that the bad event for arm $j$ also occurs at round $t+k$. Then, we have that

$$\widehat{\mu}_j^{(t+k)} \in \left[\widehat{\mu}_{(t+k)^\star}^{(t+k)} - U_{t+k} - 5\widetilde{U}_{t+k}, \widehat{\mu}_{(t+k)^\star}^{(t+k)} - U_{t+k}/2 + 3\widetilde{U}_{t+k}\right].$$

First, notice that since the concentration inequality under event $\mathcal{E}$ holds for rounds $t, t+k$ we have that $\widehat{\mu}_j^{(t+k)} \leq \widehat{\mu}_j^{(t)} + \widetilde{U}_t + \widetilde{U}_{t+k}$. Thus, combining it with the above inequalities gives us

$$\widehat{\mu}_{(t+k)^\star}^{(t+k)} - U_{t+k} - 5\widetilde{U}_{t+k} \leq \widehat{\mu}_j^{(t+k)} \leq \widehat{\mu}_j^{(t)} + \widetilde{U}_t + \widetilde{U}_{t+k} \leq \widehat{\mu}_{t^\star}^{(t)} - U_t/2 + 4\widetilde{U}_t + \widetilde{U}_{t+k}.$$

We now compare $\widehat{\mu}_{t^\star}^{(t)}, \widehat{\mu}_{(t+k)^\star}^{(t+k)}$. Let $o$ denote the optimal arm. We have that

$$\widehat{\mu}_{(t+k)^\star}^{(t+k)} \geq \widehat{\mu}_o^{(t+k)} \geq \mu_o - \widetilde{U}_{t+k} \geq \mu_{t^\star} - \widetilde{U}_{t+k} \geq \widehat{\mu}_{t^\star}^{(t)} - \widetilde{U}_t - \widetilde{U}_{t+k}.$$

This gives us that

$$\widehat{\mu}_{t^\star}^{(t)} - U_{t+k} - 6\widetilde{U}_{t+k} - \widetilde{U}_t \leq \widehat{\mu}_{(t+k)^\star}^{(t+k)} - U_{t+k} - 5\widetilde{U}_{t+k}.$$

Thus, we have established that

$$\widehat{\mu}_{t^\star}^{(t)} - U_{t+k} - 6\widetilde{U}_{t+k} - \widetilde{U}_t \leq \widehat{\mu}_{t^\star}^{(t)} - U_t/2 + 4\widetilde{U}_t + \widetilde{U}_{t+k} \implies$$
$$U_{t+k} \geq U_t/2 - 7\widetilde{U}_{t+k} - 5\widetilde{U}_t \geq U_t/2 - 12\widetilde{U}_t.$$

Since $\beta \geq 2304$, we get that $12\widetilde{U}_t \leq U_t/4$. Thus, we get that

$$U_{t+k} \geq U_t/4.$$

Notice that

$$\frac{U_{t+k}}{U_t} = \sqrt{\frac{c_t}{c_{t+k}}},$$

thus it immediately follows that

$$\frac{c_t}{c_{t+k}} \geq \frac{1}{16} \implies \frac{q^{t+1}-1}{q^{t+k+1}-1} \geq \frac{1}{16} \implies 16\left(1 - \frac{1}{q^{t+1}}\right) \geq q^k - \frac{1}{q^{t+1}} \implies$$

$$q^k \leq 16 + \frac{1}{q^{t+1}} \leq 17 \implies k \log q \leq \log 17 \implies k \leq 5,$$

when we pick $B = \log(T)$ batches. Thus, for every arm the bad event can happen at most 6 times, by taking a union bound over the $K$ arms we see that the probability that our algorithm is not replicable is at most $O(K\sqrt{1/\beta})$, so picking $\beta = \Theta(K^2/\rho^2)$ suffices to get the result.

## C   THE PROOF OF THEOREM 6

**Theorem.** *Let $T \in \mathbb{N}, \rho \in (0,1]$. There exists a $\rho$-replicable algorithm (presented in Algorithm 3) for the stochastic $d$-dimensional linear bandit problem with $K$ arms whose expected regret is*

$$\mathbf{E}[R_T] \leq C \cdot \frac{K^2}{\rho^2}\sqrt{dT\log(KT)},$$

*for some absolute numerical constant $C > 0$, and its running time is polynomial in $d, K, T$ and $1/\rho$.*

*Proof.* Let $c, C$ be the numerical constants hidden in Lemma 5, i.e., the size of the multi-set is in the interval $[cd\log(1/\delta)/\varepsilon^2, Cd\log(1/\delta)/\varepsilon^2]$. We know that the size of each batch $n_i \in [cq^i, Cq^i]$ (see Lemma 5), so by the end of the $B-1$ batch we will have less than $n_B$ pulls left. Hence, the number of batches is at most $B$.

We first define the event $\mathcal{E}$ that the estimates of all arms after the end of each batch are accurate, i.e., for every active arm $a$ at the beginning of the $i$-th batch, at the end of the batch we have that $\left|\left\langle a, \widehat{\theta}_i - \theta^\star \right\rangle\right| \leq \widetilde{\varepsilon}_i$. Since $\delta = 1/(KT^2)$ and there are at most $T$ batches and $K$ active arms in each batch, a simple union bound shows that $\mathcal{E}$ happens with probability at least $1 - 1/T$. We condition on the event $\mathcal{E}$ throughout the rest of the proof.

We now argue about the regret bound of our algorithm. We first show that any optimal arm $a^*$ will not get eliminated. Indeed, consider any sub-optimal arm $a \in [K]$ and any batch $i \in [B]$. Under the event $\mathcal{E}$ we have that

$$\langle a, \widehat{\theta}_i \rangle - \langle a^*, \widehat{\theta}_i \rangle \leq (\langle a, \theta^* \rangle + \widetilde{\varepsilon}_i) - (\langle a^*, \theta^* \rangle - \widetilde{\varepsilon}_i) < 2\widetilde{\varepsilon}_i < \varepsilon_i + \overline{\varepsilon}_i.$$

Next, we need to bound the number of times we pull some fixed suboptimal arm $a \in [K]$. We let $\Delta = \langle a^* - a, \theta^* \rangle$ denote the gap and we let $i$ be the smallest integer such that $\varepsilon_i < \Delta/4$. We claim that this arm will get eliminated by the end of batch $i$. Indeed,

$$\langle a^*, \widehat{\theta}_i \rangle - \langle a, \widehat{\theta}_i \rangle \geq (\langle a^*, \widehat{\theta}_i \rangle - \widetilde{\varepsilon}_i) - (\langle a, \widehat{\theta}_i \rangle + \widetilde{\varepsilon}_i) = \Delta - 2\widetilde{\varepsilon}_i > 4\varepsilon_i - 2\widetilde{\varepsilon}_i > \widetilde{\varepsilon}_i + \overline{\varepsilon}_i.$$

This shows that during any batch $i$, all the active arms have gap at most $4\varepsilon_{i-1}$. Thus, the regret of the algorithm conditioned on the event $\mathcal{E}$ is at most

$$\sum_{i=1}^{B} 4n_i\varepsilon_{i-1} \leq 4\beta C \sum_{i=1}^{B} q^i \sqrt{d\log(KT^2)/q^{i-1}} \leq 6\beta Cq\sqrt{d\log(KT)} \sum_{i=0}^{B-1} q^{i/2} \leq$$

$$O\left(\beta q^{B/2+1}\sqrt{d\log(KT)}\right) = O\left(\frac{K^2}{\rho^2}q^{B/2+1}\sqrt{d\log(KT)}\right) = O\left(\frac{K^2}{\rho^2}q\sqrt{dT\log(KT)}\right).$$

Thus, the overall regret is bounded by $\delta \cdot T + (1-\delta) \cdot O\left(\frac{K^2}{\rho^2}q\sqrt{dT\log(KT)}\right) = O\left(\frac{K^2}{\rho^2}q\sqrt{dT\log(KT)}\right).$

We now argue about the replicability of our algorithm. The analysis follows in a similar fashion as in Theorem 4. Let $\widehat{\theta}_i, \widehat{\theta}'_i$ be the LSE after the $i$-th batch, under two different executions of the algorithm and assume that the set of active arms. We condition on the event $\mathcal{E}'$ for the other execution as well. Assume that the set of active arms is the same under both executions at the beginning of batch $i$. Notice that since the set that is guaranteed by Lemma 5 is computed by a deterministic algorithm, both executions will pull the same arms in batch $i$. Consider a suboptimal arm $a$ and let $a_{i^*} = \arg\max_{a \in \mathcal{A}} \langle \widehat{\theta}_i, a \rangle, a'_{i^*} = \arg\max_{a \in \mathcal{A}} \langle \widehat{\theta}'_i, a \rangle$. Under the event $\mathcal{E} \cap \mathcal{E}'$ we have that $|\langle a, \widehat{\theta}_i - \widehat{\theta}'_i \rangle| \leq 2\widetilde{\varepsilon}_i, |\langle a_{i^*}, \widehat{\theta}_i - \widehat{\theta}'_i \rangle| \leq 2\widetilde{\varepsilon}_i$, and $|\langle a'_{i^*}, \widehat{\theta}'_i \rangle - \langle a_{i^*}, \widehat{\theta}_i \rangle| \leq 2\widetilde{\varepsilon}_i$. Notice that, since the randomness of $\overline{\varepsilon}_i$ is shared, if $\langle a, \widehat{\theta}_i \rangle + \widetilde{\varepsilon}_i \geq \langle a_{i^*}, \widehat{\theta}_i \rangle - \overline{\varepsilon}_i + 4\widetilde{\varepsilon}_i$, then the arm $a$ will not be eliminated after the $i$-th batch in some other execution of the algorithm as well. Similarly, if $\langle a, \widehat{\theta}_i \rangle + \widetilde{\varepsilon}_i < \langle a_{i^*}, \widehat{\theta}_i \rangle - \overline{\varepsilon}_i - 4\widetilde{\varepsilon}_i$ the the arm $a$ will get eliminated after the $i$-th batch in some other execution of the algorithm as well. In particular, this means that if $\langle a, \widehat{\theta}_i \rangle - 2\widetilde{\varepsilon}_i > \langle a_{i^*}, \widehat{\theta}_i \rangle + \widetilde{\varepsilon}_i - \varepsilon_i/2$ then the arm $a$ will not get eliminated in some other execution of the algorithm and if $\langle a, \widehat{\theta}_i \rangle + 5\widetilde{\varepsilon}_i < \langle a_{i^*}, \widehat{\theta}_i \rangle - \varepsilon_i$ then the arm $j$ will also get eliminated in some other execution of the algorithm with probability 1 under the event $\mathcal{E} \cap \mathcal{E}'$. Thus, it suffices to bound the probability that the decision about arm $j$ will be different between the two executions when we are in neither of these cases. Then, the worst case bound due to the mass of the uniform probability measure is

$$\frac{16\sqrt{d\log(1/\delta)/\widetilde{c}_i}}{\sqrt{d\log(1/\delta)/c_i}} .$$

This implies that the probability mass of the bad event is at most $16\sqrt{c_i/\widetilde{c}_i} = 16\sqrt{1/\beta}$. A naive union bound would require us to pick $\beta = \Theta(K^2 \log^2 T/\rho^2)$. We next show to avoid the $\log^2 T$ factor. Fix a sub-optimal arm $a \in [K]$ and let $t$ be the first round that it appears in the bad event.

Since the bad event occurs at round $t$, we know that

$$\langle a, \widehat{\theta}_t \rangle \in \left[ \langle a_{t^*}, \widehat{\theta}_t \rangle - \varepsilon_t - 5\widetilde{\varepsilon}_t, \langle a_{t^*}, \widehat{\theta}_t \rangle - \varepsilon_t/2 + 3\widetilde{\varepsilon}_t \right].$$

In the above, $a_{t^*}$ is the optimal arm at round $t$ w.r.t. the LSE. Now assume that the bad event for arm $a$ also occurs at round $t + k$. Then, we have that

$$\langle a, \widehat{\theta}_{t+k} \rangle \in \left[ \langle a_{(t+k)^*}, \widehat{\theta}_{t+k} \rangle - \varepsilon_{t+k} - 5\widetilde{\varepsilon}_{t+k}, \langle a_{(t+k)^*}, \widehat{\theta}_{t+k} \rangle - \varepsilon_t/2 + 3\widetilde{\varepsilon}_{t+k} \right].$$

First, notice that since the concentration inequality under event $\mathcal{E}$ holds for rounds $t, t + k$ we have that $\langle a, \widehat{\theta}_{t+k} \rangle \leq \langle a, \widehat{\theta}_t \rangle + \widetilde{\varepsilon}_t + \widetilde{\varepsilon}_{t+k}$. Thus, combining it with the above inequalities gives us

$$\langle a_{(t+k)^*}, \widehat{\theta}_{t+k} \rangle - \varepsilon_{t+k} - 5\widetilde{\varepsilon}_{t+k} \leq \langle a, \widehat{\theta}_{t+k} \rangle \leq \langle a, \widehat{\theta}_t \rangle + \widetilde{\varepsilon}_t + \widetilde{\varepsilon}_{t+k} \leq \langle a_{t^*}, \widehat{\theta}_t \rangle - \varepsilon_t/2 + 4\widetilde{\varepsilon}_t + \widetilde{\varepsilon}_{t+k}.$$

We now compare $\langle a_{t^*}, \widehat{\theta}_t \rangle, \langle a_{(t+k)^*}, \widehat{\theta}_{t+k} \rangle$. Let $a^*$ denote the optimal arm. We have that

$$\langle a_{(t+k)^*}, \widehat{\theta}_{t+k} \rangle \geq \langle a^*, \widehat{\theta}_{t+k} \rangle \geq \langle a^*, \theta^* \rangle - \widetilde{\varepsilon}_{t+k} \geq \langle a_{t^*}, \theta^* \rangle - \widetilde{\varepsilon}_{t+k} \geq \langle a_{t^*}, \widehat{\theta}_t \rangle - \widetilde{\varepsilon}_{t+k} - \widetilde{\varepsilon}_t.$$

This gives us that

$$\langle a_{t^*}, \widehat{\theta}_t \rangle - \varepsilon_{t+k} - 6\widetilde{\varepsilon}_{t+k} - \widetilde{\varepsilon}_t \leq \langle a_{(t+k)^*}, \widehat{\theta}_{t+k} \rangle - \varepsilon_{t+k} - 5\widetilde{\varepsilon}_{t+k}.$$

Thus, we have established that

$$\langle a_{t^*}, \widehat{\theta}_t \rangle - \varepsilon_{t+k} - 6\widetilde{\varepsilon}_{t+k} - \widetilde{\varepsilon}_t \leq \langle a_{t^*}, \widehat{\theta}_t \rangle - \varepsilon_t/2 + 4\widetilde{\varepsilon}_t + \widetilde{\varepsilon}_{t+k} \implies$$
$$\varepsilon_{t+k} \geq \varepsilon_t/2 - 7\widetilde{\varepsilon}_{t+k} - 5\widetilde{\varepsilon}_t \geq \varepsilon_t/2 - 12\widetilde{\varepsilon}_t.$$

Since $\beta \geq 2304$, we get that $12\widetilde{\varepsilon}_t \leq \varepsilon_t/4$. Thus, we get that

$$\varepsilon_{t+k} \geq \varepsilon_t/4.$$

Notice that

$$\frac{\varepsilon_{t+k}}{\varepsilon_t} = \sqrt{\frac{q^t}{q^{t+k}}},$$

thus it immediately follows that

$$\frac{q^t}{q^{t+k}} \geq \frac{1}{16} \implies q^k \leq 16 \implies k \log q \leq \log 16 \implies k \leq 4,$$

when we pick $B = \log(T)$ batches. Thus, for every arm the bad event can happen at most 5 times, by taking a union bound over the $K$ arms we see that the probability that our algorithm is not replicable is at most $O(K\sqrt{1/\beta})$, so picking $\beta = \Theta(K^2/\rho^2)$ suffices to get the result. $\qquad\square$

## D  Naive Application of Algorithm 3 with Infinite Action Space

We use a $1/T^{1/(4d+2)}$−net that has size at most $(3T)^{\frac{d}{4d+2}}$. Let $\mathcal{A}'$ be the new set of arms. We then run Algorithm 3 using $\mathcal{A}'$. This gives us the following result, that is proved right after.

**Corollary 12.** *Let* $T \in \mathbb{N}, \rho \in (0, 1]$. *There is a $\rho$-replicable algorithm for the stochastic d-dimensional linear bandit problem with infinite arms whose expected regret is at most*

$$\mathbf{E}[R_T] \le C \cdot \frac{T^{\frac{4d+1}{4d+2}}}{\rho^2} \sqrt{d \log(T)},$$

*where $C > 0$ is an absolute numerical constant.*

*Proof.* Since $K \le (3T)^{\frac{d}{4d+2}}$, we have that

$$T \sup_{a\in\mathcal{A}'} \langle a, \theta^* \rangle - \mathbf{E}\left[\sum_{i=1}^T \langle a_t, \theta^* \rangle\right] \le O\left(\frac{(3T)^{\frac{2d}{4d+2}}}{\rho^2} \sqrt{dT \log\left(T(3T)^{\frac{d}{4d+2}}\right)}\right) = O\left(\frac{T^{\frac{4d+1}{4d+2}}}{\rho^2} \sqrt{d \log(T)}\right)$$

Comparing to the best arm in $\mathcal{A}$, we have that:

$$T \sup_{a\in\mathcal{A}} \langle a, \theta^* \rangle - \mathbf{E}\left[\sum_{i=1}^T \langle a_t, \theta^* \rangle\right] = \left(T \sup_{a\in\mathcal{A}} \langle a, \theta^* \rangle - T \sup_{a\in\mathcal{A}'} \langle a, \theta^* \rangle\right) + \left(T \sup_{a\in\mathcal{A}'} \langle a, \theta^* \rangle - \mathbf{E}\left[\sum_{i=1}^T \langle a_t, \theta^* \rangle\right]\right)$$

Our choice of the $1/T^{1/(4d+2)}$-net implies that for every $a \in \mathcal{A}$ there exists some $a' \in \mathcal{A}'$ such that $||a - a'||_2 \le 1/T^{1/(4d+2)}$. Thus, $\sup_{a\in\mathcal{A}}\langle a, \theta^* \rangle - \sup_{a'\in\mathcal{A}'}\langle a', \theta^* \rangle \le ||a - a'||_2 ||\theta^*||_2 \le 1/T^{1/(4d+2)}$. Thus, the total regret is at most

$$T \cdot 1/T^{1/(4d+2)} + O\left(\frac{T^{\frac{4d+1}{4d+2}}}{\rho^2} \sqrt{d \log(T)}\right) = O\left(\frac{T^{\frac{4d+1}{4d+2}}}{\rho^2} \sqrt{d \log(T)}\right).$$

$\square$

## E  The Proof of Theorem 10

**Theorem.** *Let $T \in \mathbb{N}, \rho \in (0, 1]$. There exists a $\rho$-replicable algorithm (presented in Algorithm 4) for the stochastic d-dimensional linear bandit problem with infinite action set whose expected regret is*

$$\mathbf{E}[R_T] \le C \cdot \frac{d^4 \log(d) \log^2 \log(d) \log\log\log(d)}{\rho^2} \sqrt{T} \log^{3/2}(T),$$

*for some absolute numerical constant $C > 0$, and its running time is polynomial in $T^d$ and $1/\rho$.*

*Proof.* First, the algorithm is $\rho$-replicable since in each batch we use a replicable LSE sub-routine with parameter $\rho' = \rho/B$. This implies that

$$\mathbf{Pr}[(a_1, ..., a_T) \ne (a'_1, ..., a'_T)] = \mathbf{Pr}[\exists i \in [B] : \widehat{\theta}_i \text{ was not replicable}] \le \rho.$$

Let us fix a batch iteration $i \in [B-1]$. Set $\mathcal{C}_i$ be the core set computed by Lemma 7. The algorithm first pulls $n_i = \frac{Cd^4 \log(d/\delta) \log^2 \log(d) \log\log\log(d)}{\varepsilon_i^2 \rho'^2}$ times each one of the arms of the $i$-th core set $\mathcal{C}_i$, as indicated by Lemma 9 and computes the LSE $\widehat{\theta}_i$ in a replicable way using the algorithm of Lemma 9. Let $\mathcal{E}$ be the event that over all batches the estimations are correct. We pick $\delta = 1/(2|\mathcal{A}'|T^2)$ so that this good event does hold with probability at least $1 - 1/T$. Our goal is to control the expected regret which can be written as

$$\mathbf{E}[R_T] = T \sup_{a\in\mathcal{A}} \langle a, \theta^\star \rangle - \mathbf{E} \sum_{t=1}^T \langle a_t, \theta^\star \rangle.$$

We have that

$$T \sup_{a\in\mathcal{A}} \langle a, \theta^\star \rangle - T \sup_{a'\in\mathcal{A}'} \langle a', \theta^\star \rangle \le 1,$$

since $\mathcal{A}'$ is a deterministic $1/T$-net of $\mathcal{A}$. Also, let us set the expected regret of the bounded action sub-problem as

$$\mathbf{E}[R'_T] = T \sup_{a' \in \mathcal{A}'} \langle a', \theta^\star \rangle - \mathbf{E} \sum_{t=1}^{T} \langle a_t, \theta^\star \rangle \,.$$

We can now employ the analysis of the finite arm case. During batch $i$, any active arm has gap at most $4\varepsilon_{i-1}$, so the instantaneous regret in any round is not more than $4\varepsilon_{i-1}$. The expected regret conditional on the good event $\mathcal{E}$ is upper bounded by

$$\mathbf{E}[R'_T | \mathcal{E}] \le \sum_{i=1}^{B} 4 M_i \varepsilon_{i-1} \,,$$

where $M_i$ is the total number of pulls in batch $i$ (using the replicability blow-up) and $\varepsilon_{i-1}$ is the error one would achieve by drawing $q^i$ samples (ignoring the blow-up). Then, for some absolute constant $C > 0$, we have that

$$\mathbf{E}[R'_T | \mathcal{E}] \le \sum_{i=1}^{B} 4 \left( q^i \frac{d^3 \log(d) \log^2 \log(d) \log \log \log(d) \log^2 T}{\rho^2} \right) \cdot \sqrt{d^2 \log(T)/q^{i-1}} \,,$$

which yields that

$$\mathbf{E}[R'_T | \mathcal{E}] \le C \frac{d^4 \log(d) \log^2 \log(d) \log \log \log(d) \log(T) \sqrt{\log(T)}}{\rho^2} \cdot S \,,$$

where we set

$$S := \sum_{i=1}^{B} \frac{q^i}{q^{(i-1)/2}} = q^{1/2} \sum_{i=1}^{B} q^{i/2} = q^{(1+B)/2} \,.$$

We pick $B = \log(T)$ and get that, if $q = T^{1/B}$ then $S = \Theta(\sqrt{T})$. We remark that this choice of $q$ is valid since

$$\sum_{i=1}^{B} q^i = \frac{q^{B+1} - q}{q - 1} = \Theta(q^B) - 1 \ge \frac{T \rho^2}{d^3 \log(d) \log^2 \log(d) \log \log \log(d)} \,.$$

Hence, we have that

$$\mathbf{E}[R'_T | \mathcal{E}] \le O \left( \frac{d^4 \log(d) \log^2 \log(d) \log \log \log(d)}{\rho^2} \sqrt{T} \log^{3/2}(T) \right) \,.$$

Note that when $\mathcal{E}$ does not hold, we can bound the expected regret by $1/T \cdot T = 1$. This implies that the overall regret $\mathbf{E}[R_T] \le 2 + \mathbf{E}[R'_T | \mathcal{E}]$ and so it satisfies the desired bound and the proof is complete. $\qquad \square$

## F   DEFERRED LEMMATA

### F.1   THE PROOF OF LEMMA 8

*Proof.* Consider the distribution $\pi$ that is a 2-approximation to the optimal G-design and has support $|\mathcal{C}| = O(d \log \log d)$. Let $\mathcal{C}'$ be the set of arms in the support such that $\pi(a) \le c/d \log d$. We consider $\widetilde{\pi} = (1 - x)\pi + xa$, where $a \in \mathcal{C}'$ and $x$ will be specified later. Consider now the matrix $V(\widetilde{\pi})$. Using the Sherman-Morrison formula, we have that

$$V(\widetilde{\pi})^{-1} = \frac{1}{1-x} V(\pi)^{-1} - \frac{x V(\pi)^{-1} a a^\top V(\pi)^{-1}}{(1-x)^2 \left( 1 + \frac{1}{1-x} ||a||^2_{V(\pi)^{-1}} \right)} = \frac{1}{1-x} \left( V(\pi)^{-1} - \frac{x V(\pi)^{-1} a a^\top V(\pi)^{-1}}{1 - x + ||a||^2_{V(\pi)^{-1}}} \right) \,.$$

Consider any arm $a'$. Then,

$$||a'||^2_{V(\widetilde{\pi})^{-1}} = \frac{1}{1-x} ||a||^2_{V(\pi)^{-1}} - \frac{x}{1-x} \cdot \frac{(a^\top V(\pi)^{-1} a')^2}{1 - x + ||a||^2_{V(\pi)^{-1}}} \le \frac{1}{1-x} ||a||^2_{V(\pi)^{-1}} \,.$$

Note that we apply this transformation at most $O(d \log \log d)$ times. Let $\widehat{\pi}$ be the distribution we end up with. We see that

$$||a'||^2_{V(\widehat{\pi})^{-1}} \leq \left(\frac{1}{1-x}\right)^{cd \log \log d} ||a||^2_{V(\pi)^{-1}} \leq 2 \left(\frac{1}{1-x}\right)^{cd \log \log d} d.$$

Notice that there is a constant $c'$ such that when $x = c'/d \log d$ we have that $\left(\frac{1}{1-x}\right)^{cd \log \log d} \leq 2$. Moreover, notice that the mass of every arm is at least $x(1-x)^{|\mathcal{C}|} \geq x - |\mathcal{C}|x^2 = c'/(d\log(d)) - c''d \log \log d/(d^2 \log^2(d)) \geq c/(d \log(d))$, for some absolute numerical constant $c > 0$. This concludes the claim. $\qquad\square$

## F.2 THE PROOF OF LEMMA 9

*Proof.* The proof works when we can treat $\Omega(\lceil d \log(1/\delta)\pi(a)/\varepsilon^2\rceil)$ as $\Omega(d \log(1/\delta)\pi(a)/\varepsilon^2)$, i.e., as long as $\pi(a) = \Omega(\varepsilon^2/d \log(1/\delta))$. In the regime we are in, this point is handled thanks to Lemma 8. Combining the following proof with Lemma 8, we can obtain the desired result.

We underline that we work in the fixed design setting: the arms $a_i$ are deterministically chosen independently of the rewards $r_i$. Assume that the core set of Lemma 7 is the set $\mathcal{C}$. Fix the multi-set $S = \{(a_i, r_i) : i \in [M]\}$, where each arm $a$ lies in the core set and is pulled $n_a = \Theta(\pi(a)d \log(d) \log(|\mathcal{C}|/\delta)/\varepsilon^2)$ times[2]. Hence, we have that

$$M = \sum_{a \in \mathcal{C}} n_a = \Theta\left(d \log(d) \log(|\mathcal{C}|/\delta)/\varepsilon^2\right).$$

Let also $V = \sum_{i \in [M]} a_i a_i^\top$. The least-squares estimator can be written as

$$\theta_{\text{LSE}}^{(\varepsilon)} = V^{-1} \sum_{i \in [M]} a_i r_i = V^{-1} \sum_{a \in \mathcal{C}} a \sum_{i \in [n_a]} r_i(a),$$

where each $a$ lies in the core set (deterministically) and $r_i(a)$ is the $i$-th reward generated independently by the linear regression process $\langle \theta^\star, a \rangle + \xi$, where $\xi$ is a fresh zero mean sub-gaussian random variable. Our goal is to reproducibly estimate the value $\sum_{i \in [n_a]} r_i(a)$ for any $a$. This is sufficient since two independent executions of the algorithm share the set $\mathcal{C}$ and $n_a$ for any $a$. Note that the above sum is a random variable. In the following, we condition on the high-probability event that the average reward of the arm $a$ is $\varepsilon$-close to the expected one, i.e., the value $\langle \theta^\star, a \rangle$. This happens with probability at least $1 - \delta/(2|\mathcal{C}|)$, given $\Omega(\pi(a)d \log(d) \log(|\mathcal{C}|/\delta)/\varepsilon^2)$ samples from arm $a \in \mathcal{C}$. In order to guarantee replicability, we will apply a result from Impagliazzo et al. (2022). Since we will union bound over all arms in the core set and $|\mathcal{C}| = O(d \log \log(d))$ (via Lemma 7), we will make use of a $(\rho/|\mathcal{C}|)$-replicable algorithm that gives an estimate $v(a) \in \mathbb{R}$ such that

$$|\langle \theta^\star, a \rangle - v(a)| \leq \tau,$$

with probability at least $1 - \delta/(2|\mathcal{C}|)$. For $\delta < \rho$, the algorithm uses

$$S_a = \Omega\left(d^2 \log(d/\delta) \log^2 \log(d) \log \log \log(d)/(\rho^2 \tau^2)\right)$$

many samples from the linear regression with fixed arm $a \in \mathcal{C}$. Since we have conditioned on the randomness of $r_i(a)$ for any $i$, we get

$$\left|\frac{1}{n_a} \sum_{i \in [n_a]} r_i(a) - v(a)\right| \leq \left|\frac{1}{n_a} \sum_{i \in [n_a]} r_i(a) - \langle \theta^*, a \rangle\right| + |\langle \theta^*, a \rangle - v(a)| \leq \varepsilon + \tau,$$

with probability at least $1 - \delta/(2|\mathcal{C}|)$. Hence, by repeating this approach for all arms in the core set, we set $\theta_{\text{SQ}} = V^{-1} \sum_{a \in \mathcal{C}} a\, n_a\, v(a)$. Let us condition on the randomness of the estimate $\theta_{\text{LSE}}^{(\varepsilon)}$. We have that

$$\sup_{a' \in \mathcal{A}} |\langle a', \theta_{\text{SQ}} - \theta^\star \rangle| \leq \sup_{a' \in \mathcal{A}} |\langle a', \theta_{\text{SQ}} - \theta_{\text{LSE}}^{(\varepsilon)}\rangle| + \sup_{a' \in \mathcal{A}} |\langle a', \theta_{\text{LSE}}^{(\varepsilon)} - \theta^\star\rangle|.$$

---

[2]Recall that $\pi(a) \geq c/(d \log(d))$, for some constant $c > 0$, so the previous expression is $\Omega(\log(\delta/|\mathcal{C}|)/\varepsilon^2)$.

Note that the second term is $\varepsilon$ with probability at least $1 - \delta$ via Lemma 5. Our next goal is to tune the accuracy $\tau \in (0, 1)$ so that the first term yields another $\varepsilon$ error. For the first term, we have that

$$\sup_{a' \in \mathcal{A}} |\langle a', \theta_{\text{SQ}} - \theta_{\text{LSE}}^{(\varepsilon)} \rangle| \leq \sup_{a' \in \mathcal{A}} \left| \langle a', V^{-1} \sum_{a \in \mathcal{C}} a \, n_a \, (\varepsilon + \tau) \rangle \right|$$

Note that $V = \frac{Cd \log(d) \log(|\mathcal{C}|/\delta)}{\varepsilon^2} \sum_{a \in \mathcal{C}} \pi(a) a a^\top$ and so $V^{-1} = \frac{\varepsilon^2}{Cd \log(d) \log(|\mathcal{C}|/\delta)} V(\pi)^{-1}$, for some absolute constant $C > 0$. This implies that

$$\sup_{a' \in \mathcal{A}} |\langle a', \theta_{\text{SQ}} - \theta_{\text{LSE}}^{(\varepsilon)} \rangle| \leq (\varepsilon + \tau) \sup_{a' \in \mathcal{A}} \left| \left\langle a', \frac{\varepsilon^2}{Cd \log(d) \log(|\mathcal{C}|/\delta)} V(\pi)^{-1} \sum_{a \in \mathcal{C}} \frac{Cd \log(d) \log(|\mathcal{C}|/\delta) \pi(a)}{\varepsilon^2} a \right\rangle \right| .$$

Hence, we get that

$$\sup_{a' \in \mathcal{A}} |\langle a', \theta_{\text{SQ}} - \theta_{\text{LSE}}^{(\varepsilon)} \rangle| \leq (\varepsilon + \tau) \sup_{a' \in \mathcal{A}} \left| \left\langle a', V(\pi)^{-1} \sum_{a \in \mathcal{C}} \pi(a) a \right\rangle \right| .$$

Consider a fixed arm $a' \in \mathcal{A}$. Then,

$$\left| \left\langle a', V(\pi)^{-1} \sum_{a \in \mathcal{C}} \pi(a) a \right\rangle \right| \leq \sum_{a \in \mathcal{C}} \pi(a) \left| \langle a', V(\pi)^{-1} a \rangle \right|$$

$$\leq \sum_{a \in \mathcal{C}} \pi(a) \left( 1 + \left| \langle a', V(\pi)^{-1} a \rangle \right|^2 \right)$$

$$= 1 + \sum_{a \in \mathcal{C}} \pi(a) \left| \langle a', V(\pi)^{-1} a \rangle \right|^2$$

$$= 1 + ||a'||_{V(\pi)^{-1}}^2$$

$$\leq 4d + 1 ,$$

where the last inequality follows from the fact that $\pi$ is a 4-approximation of the $G$-optimal design. Hence, in total, by picking $\tau = \varepsilon$, we get that

$$\sup_{a' \in \mathcal{A}} |\langle a', \theta_{\text{SQ}} - \theta^\star \rangle| \leq 11 d \varepsilon .$$

Thus, for any $\varepsilon > 0$, the total number of pulls of each arm is

$$\Omega \left( d^4 \log(d/\delta) \log^2 \log(d) \log \log \log(d)/(\rho^2 \varepsilon^2) \right) ,$$

to get

$$\sup_{a' \in \mathcal{A}} |\langle a', \theta_{\text{SQ}} - \theta^\star \rangle| \leq \varepsilon .$$

$\square$

# G    COMPUTATIONAL PERFORMANCE OF ALGORITHM 4

In this appendix, we discuss the barriers towards computational efficiency regarding Algorithm 4. The reasons why Algorithm 4 is computationally inefficient are the following: (a) we have to compute the arm in the set of active arms that has maximum correlation with the estimate $\widehat{\theta}_i$, (b) we have to eliminate arms based on this value and (c) we have to run at each batch the Frank-Wolfe algorithm (or some other optimization method needed for Lemma 5) in order to obtain an approximate G-optimal design. As a minimal assumption in what follows, we focus on the case where the action set $\mathcal{A}$ is convex and we have access to a separation oracle for it.

Note that executing both (a) and (b) naively requires time exponential in $d$. However, on the one side arm elimination (issue (b)) reduces to finding the intersection of the current active set with a halfspace $\mathcal{H}$ whose normal vector is $\widehat{\theta}^i$ and the threshold is, roughly speaking, the maximum correlation. This maximum correlation can also be computed efficiently. Finding an arm with (almost) maximum correlation relates to the problem of finding a point that maximizes a linear

objective under the constraint that the point lies in the intersection of the active arm set with some linear constraints. Thus, we can use the ellipsoid algorithm to implement this step.

The above discussion deals with issues (a) and (b) and, essentially, states that even with infinitely many actions, one could implement these steps efficiently. We now focus on issue (c). The Frank-Wolfe method first requires a proper initialization. As mentioned in Lattimore & Szepesvári (2020), if the starting point is chosen to be the uniform distribution over $\mathcal{A}'$, then the number of iterations before getting a 2-approximate optimal design is roughly $\widetilde{O}(d)$. The issue is that since $\mathcal{A}'$ is exponential in $d$, it is not clear how to work with such an initialization efficiently. Notably there is a different initialization Fedorov (2013); Lattimore et al. (2020) with support $O(d)$ for which the method runs in $O(d \log \log(d))$ rounds (see Note 3 at Section 21.2 of Lattimore & Szepesvári (2020) and Lattimore et al. (2020)). There are two issues: first, one requires an oracle to provide this good initialization. Second, each iteration of the Frank-Wolfe method (with current design guess $\pi$) requires computing a point in the current active set with maximum $V(\pi)^{-1}$-norm. As noted in Todd (2016), a good initialization for finding a G-optimal design, i.e., a minimum volume enclosing ellipsoid (MVEE) should be sufficiently sparse (compared to the number of active arms) and assign positive mass to arms that correspond to extreme points, i.e., points that are close to the border of MVEE. The work of Kumar & Yildirim (2005) provides an initial core set that depends only on $d$ but not on the number of points. The algorithm works as follows: it runs for $d$ iterations and, in each round, it adds 2 arms into the core set. Initially, we set the core set $\mathcal{C}_0 = \emptyset$ and let $\Psi = \{0\}$. In each iteration $i \in [d]$, the algorithm draws a random direction $v_i$ in the orthogonal complement of $\Psi$ (this step is replicable thanks to the shared randomness) and computes the vectors in the active arms' set with the maximum and the minimum correlation with $v_i$, say $a_i^+, a_i^-$. It then extends $\mathcal{C}_0 \leftarrow \mathcal{C}_0 \cup \{a_i^+, a_i^-\}$ and sets $\Psi \leftarrow \mathrm{span}(\Psi, \{a_i^+ - a_i^-\})$. Hence, the runtime of this algorithm corresponds to the runtime of the tasks $\max_{a \in \mathcal{A}'} \langle a, v_i \rangle$ and $\min_{a \in \mathcal{A}'} \langle a, v_i \rangle$. One can efficiently approximate these values using the ellipsoid algorithm and hence efficiently initialize the Frank-Wolfe algorithm as in Todd (2016) (e.g., set the weights uniformly $1/(2d)$).

Our second challenge deals with finding a point in the active arm set with maximum $V(\pi)^{-1}$-norm for some current guess $\pi$. Even if the current active set is a polytope, finding an exact norm maximizer is NP-hard Freund & Orlin (1985); Mangasarian & Shiau (1986)[3]. Hence, one should focus on efficient approximation algorithms. We note that even a $\mathrm{poly}(d)$-approximate maximizer is sufficient to get $\widetilde{O}(\mathrm{poly}(d)\sqrt{T})$ regret. Such an algorithm for polytopes, which gets an $1/d^2$-approximation, is provided in Ye (1992); Vavasis (1993).

As a general note, if we assume that we have access to an oracle $\mathcal{O}$ that computes a 2-approximate G-optimal design in time $T_{\mathcal{O}}$, then our Algorithm 4 runs in time polynomial in $T_{\mathcal{O}}$.

## H  EXPERIMENTAL EVALUATION

In this section, we provide some experimental evaluation of our proposed algorithms in the setting of multi-armed stochastic bandits. In particular, we will compare the performance of our algorithm with the standard UCB approach. We first analyze the experimental setup.

**Experimental Setup.**  We consider a multi-armed bandit setting with $K = 6$ arms that have Bernoulli rewards with bias $(0.44, 0.47, 0.5, 0.53, 0.56, 0.59)$, we run the algorithms for $T = 45000$ iterations and we execute both algorithms 20 different times. For our algorithm, we set its replicability parameter $\rho = 0.3$. We are interested in comparing the replicability and the regret of the two algorithms. We underline that our theoretical bound in this setting shows that the regret of our algorithm could be, roughly, at most 360 times worse than the regret of UCB.

**Observations.**  We first observe that our proposed bandit algorithm is indeed replicable, i.e., it satisfies our proposed reproducibility/replicability definition. In particular, for the 10 pairs of consecutive executions we consider, we observe that our algorithm pulls the exact same sequence of arms in all of them. On the other hand, the standard UCB algorithm is far from being replicable. To be more precise, between consecutive executions we see that the algorithm makes a different choice

---

[3]In fact, even finding a constant factor approximation, for some appropriate constant, is NP-hard Bellare & Rogaway (1993).

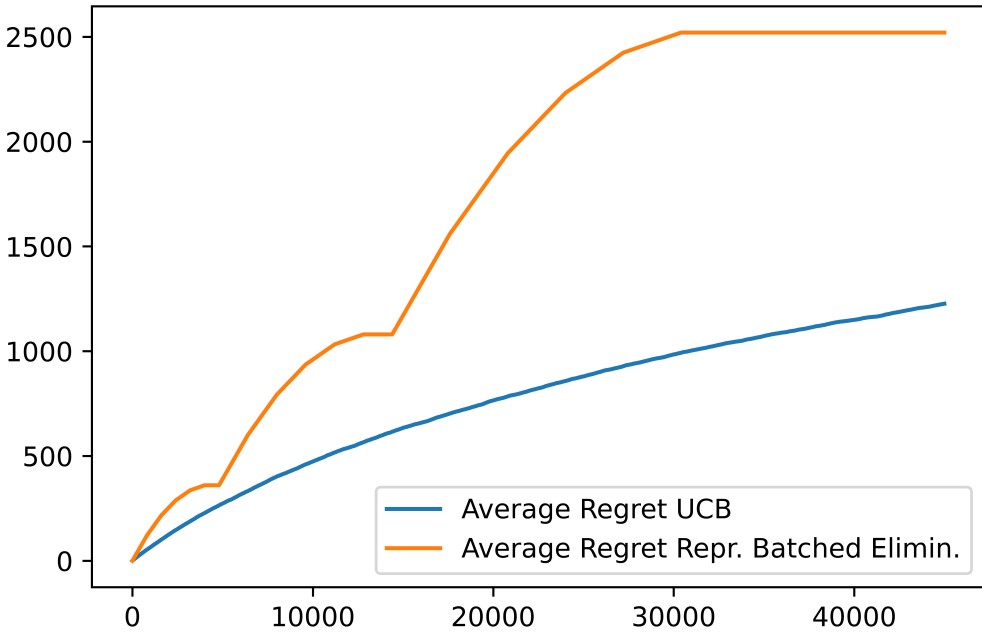

Figure 1: Cumulative Regret UCB vs. Replicable Batched Algorithm

between $26208$ to $28002$ rounds out of the $45000$ rounds, which is more that half of the rounds! However, the regret of our approach is quite competitive compared to the standard UCB algorithm. In particular, the regret of the standard UCB algorithm has the expected polylog($T$) shape, while the regret of our replicable variant incurs a multiplicative overhead that is, roughly, $2.5$ times worse than UCB. This is depicted in Figure 1, where we plot the average cumulative regret (across the $20$ executions) of the two algorithms.

