# OpenReview forum: "Replicable Bandits"
_ICLR.cc/2023/Conference — ICLR 2023 poster_

### Official Review · Reviewer_4C9J · 2022-10-13

**Confidence:** 4
**Correctness:** 4
**Technical Novelty And Significance:** 2
**Empirical Novelty And Significance:** Not applicable
**Recommendation:** 6

**Clarity, Quality, Novelty And Reproducibility:**

Clarity

The section 2.1 is not very helpful for understanding. In fact, after I read this part, I guess there is a $\log{1\over \rho}$ factor in the regret upper bound, while it should be ${1\over \rho^2}$.

I suggest the authors to give some insights about the ${1\over \rho^2}$ factor, so that the readers could understand the real hardness of this problem. For example, in section 2.2, the authors could explain about where the ${1\over \rho^2}$ factor in the complexity of ReprMeanEstimation comes from (i.e., from the random offset used in ReprMeanEstimation). After these explanations, one can understand the reason of using random $\bar{U}_i$ (or $\bar{\epsilon}_i$) in Algorithm 2 and 3 as well.

Some other minor typos.

- In the last sentence of the first paragraph, there are two "to the".
- In related works, the regret upper bound in (Bubeck et al., 2012a) when there are $K$ arms is $O(\sqrt{dT\log K})$, but not $O(d\sqrt{T\log K})$.
- In Proposition 2, why using $\Omega$ but not $O$?


**Strength And Weaknesses:**

Strength

1. Applying reproducity to bandits problems in an interesting and novel idea.

2. I check some of the proofs in appendix, and they seem to be correct.

Weaknesses

1. About the motivation.

Though the reproducity of scientific findings are very important, I do not really understand why we need to design reproducible algorithms for bandit problems. Maybe one can claim that we can reproduce others' experimental results easily when the designed algorithms are reproducible, but in this case, I think i) the definition of reproducible (Definition 1) is too strict; and ii) the cost is too large. And  I guess that ii) maybe a consequence of i). In fact, to reproduce others' experimental results, we do not really need the sequence to be exact the same.

2. About the regret upper bounds.

The regret upper bound seems to be not very tight. For example, when we choose $\rho \to 1$, the algorithms does not have any reproducible properties, but we still suffer an extra $K^2$ factor in the regret upper bound. What is the reason of this?

3. Lack of regret lower bounds.

I guess that the regret lower bound analysis could be difficult, especially for the linear bandit case. However, I think some discussions would be very helpful here. After I read the paper, I cannot really understand the difficulty of designing reproducible algorithms, and also do not know which kinds of instances could be very hard to solve in this case. Besides, it is mentioned in the comclusion that the factor of ${1\over \rho^2}$ is tight (according to (Impagliazzo et al., 2022)). However, after I read that paper, I do not think its proof can be directly used for the bandit case. Maybe I miss some important things, but I think there should be more detailed explanations.

**Summary Of The Paper:**

This paper designs reproducible policies for classic multi-armed bandit model and linear bandit model. In the classic multi-armed bandit setting, the authors first design an algorithm that directly applies existing reproducible algorithms for estimating the means of random distributions, and propose an regret upper bound that is  $O({K^2\log^2 T \over \rho^2})$ larger than non-reproducible algorithms. Here $K$ is the number of arms, $T$ is the time horizon, and $\rho$ is the reproducible rate (i.e., with probability at least $1-\rho$, the algorithm will output the same sequence). Then they improve their algorithm by some specific properties of bandits setting, and reduce the $O(\log^2 T)$ factor in the regret upper bound. As for the linear bandit case, the authors first consider the finite arm set case, and propose an algorithm that achieves regret upper bound of $\tilde{O}({K^2 \over \rho^2}\sqrt{dT})$, which is also $\tilde{O}({K^2 \over \rho^2})$ higher than existing non-reproducible algorithms. When the arm set is infinite, the authors also provide a reproducible algorithm that achieves $\tilde{O}({d^3 \over \rho^2})$ higher regret than non-reproducible ones.

**Summary Of The Review:**

My main concern here is about the motivation of this paper, and therefore I give the score of "marginally below the threshold".

=======After the rebuttal===========

After reading the explanations about the motivation and the lower bound, I am willing to change my score to 6.

---

> ### Author Response · Authors · 2022-11-17
> **Response to Reviewer 4C9J**
>
> We would like to thank the reviewer for finding the application of reproducibility in the context of bandits interesting and novel. We now comment on all the points the reviewer has raised.
>
> >*About the motivation.
> Though the reproducity of scientific findings are very important, I do not really understand why we need to design reproducible algorithms for bandit problems. Maybe one can claim that we can reproduce others' experimental results easily when the designed algorithms are reproducible, but in this case, I think i) the definition of reproducible (Definition 1) is too strict; and ii) the cost is too large. And I guess that ii) maybe a consequence of i). In fact, to reproduce others' experimental results, we do not really need the sequence to be exact the same.*
>
> Our goal in this work is to provide a definition of reproducibility in interactive decision making settings that, on the one hand, provides strong reproducibility guarantees and, on the other hand, allows us to derive non-trivial regret bounds. Inspired by the work of Impagliazzo et al. (STOC’ 21), one can view our effort as a common approach in computer science where one tries to come up with a worst-case definition, see what are the optimal guarantees under this definition, and then relax it to handle beyond worst-case settings. For instance, we can define partial reproducibility with respect to a subset of the arms $K’$, meaning that the decision whether we de-activate these arms or not should be the same between the two executions. By inspecting our proof, we can see that this will lead to a $(K’)^2$ overhead instead of $K^2$. We also note that in the linear bandit setting, if the arms lie in a low-dimensional space, we have no explicit dependence on $K$ and we can handle a, potentially, infinite number of them. Another interesting direction would be to see how we can get an instance-dependent reproducibility overhead, i.e., instead of having an extra factor that depends on $K$ to have an extra factor that depends on $\sum_{i=1}^K \Delta_i$. More to that, the fact that the reproducibility overhead in the regret bound does not increase with respect to $T$ is interesting, since one could imagine that as $T$ increases there are more chances that the two trajectories can differ so we would need to sacrifice performance (i.e., regret) in order to guarantee reproducibility. The main intuition for the absence of $T$ is that, as the number of rounds increases, the bandit algorithm has sufficiently explored the environment and hence is sufficiently stable in its decisions. To summarize, we hope and believe that our work serves as a non-trivial baseline for reproducibility in nteractive learning settings that can inspire future research with, potentially weaker, reproducibility definitions.
>
>
> Another benefit of our definition is that it makes the behavior of these algorithms more stable, interpretable, and predictable. Very often algorithms whose behavior is random make the analysis of the performance bounds tighter. For MAB, indeed, their behavior is random since it is heavily dependent on the sequence of rewards they get. However, from a practical point of view, policy makers are reluctant to use algorithms whose behavior depends heavily on randomness. We view our definition of reproducibility as a way to create decision-making algorithms whose random behavior is, in some sense, more predictable and interpretable. Notice that, for fixed internal randomness, the sequence of the pulls of the arms is distribution-dependent and not sample-dependent. In many sensitive applications like medical trials design, such properties of the underlying algorithms are crucial.
>
> In the new version of our manuscript, we have added some experimental results for a simple MAB problem in Appendix H, which show that our algorithms could behave better in practice compared to the worst-case bounds we prove.

---

> > ### Author Response · Authors · 2022-11-17
> > **Response to Reviewer 4C9J (cont.)**
> >
> > >*About the regret upper bounds.
> > The regret upper bound seems to be not very tight. For example, when we choose $\rho \rightarrow 0$, the algorithms do not have any reproducible properties, but we still suffer an extra $K^2$ factor in the regret upper bound. What is the reason of this?*
> >
> > We believe that the \emph{current} analysis of our upper bound is tight, however, it is not clear to us that the lower bound should suffer from a $K^2$ dependence. This is, indeed, a very interesting problem and we defer the optimal dependence on $K$ to future work. We believe that it could be possible to get a better bound. Notice that in our work the extra regret is, essentially, due to the parameter $\beta$ (Algorithm 2, line 3). Roughly speaking, we need to blow up the number of times we pull every arm by a sufficiently large factor that will guarantee that the union bound of the “bad” events over all the arms has a sufficiently small probability. Using our current approach, we cannot find correlations of these events across different arms that will enable us to get rid of the $K^2$ factor. If one does not care about reproducibility guarantees they can set $\beta$ equal to some constant and derive the optimal regret guarantees in this setting.
> >
> > >*Lack of regret lower bounds.
> > I guess that the regret lower bound analysis could be difficult, especially for the linear bandit case. However, I think some discussions would be very helpful here. After I read the paper, I cannot really understand the difficulty of designing reproducible algorithms, and also do not know which kinds of instances could be very hard to solve in this case. Besides, it is mentioned in the conclusion that the factor of $1/\rho^2$ is tight (according to (Impagliazzo et al., 2022)). However, after I read that paper, I do not think its proof can be directly used for the bandit case. Maybe I miss some important things, but I think there should be more detailed explanations.*
> >
> > As we mention in the conclusion, there is no reproducible algorithm that achieves instance-dependent logarithmic regret whose scaling with respect to the reproducibility parameter is $1/\rho^{2 - \varepsilon}$, for some $\varepsilon > 0$ . This follows from the lower bound that is established in Impagliazzo et al. (2021) about the number of tosses that we need to decide if a biased coin has bias $½ + c$ or $½-c$, for some small constant $c > 0$. They show that any $\rho$-reproducible algorithm needs at least $\Omega(1/\rho^2)$ samples to determine that. Let us now sketch how we can utilize this result to show our claim. We set up a bandit problem with two arms where the reward of the first arm comes from a fair coin and the reward of the second arm comes from the biased coin. Assume that there is an algorithm $\mathcal{A}$ that achieves instance-dependendent regret $O(1/\rho^{2-\varepsilon})$, for some $\varepsilon > 0.$ We run this algorithm for $T = 1/\rho^{2-\varepsilon/2}$ iterations. Notice that the algorithm will play the best arm the majority of the times, otherwise the regret will be $ \Omega(1/\rho^{2-\varepsilon/2})$, but we know that its regret is at most $O(\log(1/\rho^{2 - \varepsilon/2})/\rho^{2-\varepsilon}) < 1/\rho^{2-\varepsilon/2}.$ Since the algorithm is reproducible, this leads to a reproducible algorithm that determines the bias of the coin with less than $1/\rho^2$ flips, which is a contradiction. Hence, we can see that the $1/\rho^2$ blow-up in the regret stems from the difficulty of reproducibility deciding the bias of a coin.
> >
> > Thank you for pointing out that it is not straightforward to see the connection with the previous paper, we agree that it is better to state it more formally. We will add the formal proof to the next version of our manuscript.

---

> > > ### Comment · Reviewer_4C9J · 2022-11-23
> > > **Reply to the rebuttal**
> > >
> > > After the explanations in the rebuttal, I understand a little about the lower bound, and also why the reproducible algorithms must suffer a much higher regret. The reason is that when there are a small number of observations (and small regret), the reproducible algorithms cannot change their performance too much when we shift the expected reward of the second arm from $\mu_1 - \Delta$ to $\mu_1 + \Delta$.
> > >
> > > I think the authors should include more discussions about i) the motivation for the definition of reproducible algorithms in this paper, and ii) why reproducible algorithms must suffer from higher regret bound (e.g., $\Omega(1/\rho^2)$) in their final version.
> > >
> > > I will change my score to 6.

---

### Official Review · Reviewer_p4Aa · 2022-10-23

**Confidence:** 4
**Correctness:** 4
**Technical Novelty And Significance:** 4
**Empirical Novelty And Significance:** Not applicable
**Recommendation:** 8

**Clarity, Quality, Novelty And Reproducibility:**

The paper is well-written. The novelties provide a new perspective and algorithms for studying bandit problems.

**Strength And Weaknesses:**

- (a) Introducing reproducibility to bandits provides a new perspective to contemplate the bandit problem.
- (b) The submission presents its algorithms step-by-step, providing a clear picture of which parts are the novelties and how a novel design is combined with existing techniques to achieve the desired regret bound. The critical components of leveraging batch bandits, uniformly sampled thresholds (step 13 of Algorithm2 and step 13 of Algorithm 3), G-optimal design, LSE, and ReproducibleLSE become intuitive and easy to follow.
- (c) The submission clearly explains the challenges and the intuitions of algorithm design. Regarding similar approaches, the submission carefully compares differences between the proposed method and published ones.
- (d) The regret bounds offer insights and the impact of reproducible learning. The submission explains how to spend additional rounds to guarantee reproducibility and the extra factors compared to the conventional bounds.

Given the above, the results and contribution of this submission are substantial. The proofs are rigorous.

- (e) The minor question is, in step 9 of Algorithm 1, is a "min" missing in the assignment of \tau?

**Summary Of The Paper:**

The submission studies the intersection of reproducibility and bandits. The submission proposes and analyses four algorithms for the stochastic MABs and stochastic linear bandits, with optimal dependency on the reproducibility parameter. The submission reveals a valuable insight between reproducible interpretability and exploration-exploitation tradeoff.

**Summary Of The Review:**

Reproducibility is becoming crucial in comparing, interpreting, and understanding different methods. Introducing reproducible bandit algorithms gives us a perspective to make better sense of bandits. The reproducible bandit algorithms also link themself to the field of reproducible learning. Given these merits and the strengths listed above, I would like to recommend an acceptance of the submission.

---

> ### Author Response · Authors · 2022-11-17
> **Response to Reviewer p4Aa**
>
> We would like to thank the reviewer for finding our paper well-written, novel, and insightful. There is indeed a typo in step 9 of algorithm 1, we will fix it. Thank you for pointing it out.

---

### Official Review · Reviewer_171s · 2022-10-23

**Confidence:** 3
**Correctness:** 4
**Technical Novelty And Significance:** 3
**Empirical Novelty And Significance:** 2
**Recommendation:** 6

**Clarity, Quality, Novelty And Reproducibility:**

Clarity and quality: The paper is well written. In particular, I appreciate a lot the warm-up part in Section 2, which provided a more comprehensible view of the problem to readers.

Novelty: The topic is new and important. The algorithms inspired a lot from Esfandiari et al. (2021) which reduces a bit the novelty, but it's ok.

Reproducibility: The reproducibility aspect is not really applicable here since the paper seems to be pure theoretical, which is somehow disappointing since the paper itself is discussing about reproducibility.

**Strength And Weaknesses:**

Strength:
- The topic is novel and could have an impact on the community.
- A first definition of reproducible bandit algorithms is provided and is supported by some non-trivial algorithms.
- A very clear thinking process on how the algorithms (and improvements) are proposed.
- Correct theoretical contributions: regret bounds provided are non-trivial.

Weakness:
- One of my major concerns is that the paper didn't provide any experimental illustration while the paper is talking about reproducibility. I do understand that the authors want to provide a fundamental study of the problem, but reproducibility itself is a very practical-oriented notion for which lack of experiments seems not quite reasonable to me.
- Another related point, not necessarily a weak point but rather something arguable, is the extra dependence of $K$ in the regret bound compared to non-reproducible algorithms. It seems that the extra $K^2/\rho^2$ factor is rather difficult to be get rid of under current context. This could particularly cause problems when we have a large arm space in practice. And somehow, reproducibility makes more sense in a large scale environment. This is a question rather about the validity of the definition itself in my opinion: does it really make sense in practice? I think the authors could probably provide more discussion on that rather than (alongside some experiments ideally) stacking theoretical results for different problem settings one by one.

**Summary Of The Paper:**

The purpose of this paper is to investigate reproducible bandit algorithms while keeping a reasonable regret guarantee. The paper starts with a notion of reproducibility for a bandit algorithm following the definition of Impagliazzo et al. (2022). It states that a bandit algorithm is reproducible if two different executions of the algorithm would lead to the same sequence of played arms with high probability.

The authors first studied the problem for standard stochastic bandits in Section 3 and proposed a first version of algorithm based on reproducible mean estimation (Impagliazzo et al., 2022) and a batched bandits algorithm (Esfandiari et al., 2021) (using traditional optimal algorithms like UCB is unlikely to be feasible under current definition). This first version incurs an extra factor of $O(K^2\log^2(T)/\rho^2)$ compared to its non-reproducible counterpart. The authors then managed to get rid of a $\log^2(T)$ factor in Section 4, which reduces the extra factor to $O(K^2/\rho^2)$.

The authors also managed to propose an algorithm with the same extra $O(K^2/\rho^2)$ factor for linear bandits (finitely-armed) proposed in Section 5.1. A study in the infinitely-armed case is also provided (with an extra $d^2$ factor against its non-reproducible counterpart).

**Summary Of The Review:**

In summary, I appreciate the authors' effort and I think the topic of reproducibility is important to the community. The algorithms and regret bounds provided are rather solid under the current definition of reproducibility. But I do have some doubt on whether the definition really makes sense given the dependence of $K$ in the proposed algorithms and the lack of experimental illustration.

Misc:
- Section 1, Paragraph 1: "due to the to the" -> "due to the"

__________________________________________________________________

After rebuttal, paper solid in theoretical aspects. Some experimental illustration added, which is appreciated, though still moderate for a paper on reproducibility. I decide to increase my score.

---

> ### Author Response · Authors · 2022-11-17
> **Response to Reviewer 171s**
>
> We would like to thank the reviewer for finding our work novel and potentially impactful, our thinking process clear and our algorithms and their analyses non-trivial.
>
> >*One of my major concerns is that the paper didn't provide any experimental illustration while the paper is talking about reproducibility. I do understand that the authors want to provide a fundamental study of the problem, but reproducibility itself is a very practical-oriented notion for which lack of experiments seems not quite reasonable to me.*
>
> As the reviewer correctly points out, the main focus of our work is a mathematical treatment of a notion of reproducibility in the context of bandits. The algorithms that we propose are fairly simple and are not harder to implement compared to their non-reproducible counterparts. We have executed some experiments using synthetic data that validate our theoretical results and are illustrated in Appendix H. To be more precise, we consider a setting with $K = 6$ arms whose means are $(0.44,0.47,0.50,0.53,0.56,0.59)$, we also set $T = 45000$ and the reproducibility parameter $\rho = 0.3$. We compare the performance of UCB and our reproducible algorithm in this setting in terms of the cumulative regret and the reproducibility by executing each algorithm $20$ times in this setting. We observe that UCB is highly unstable, since between two consecutive executions it can make a different decision about which arm to pull in $26000$ out of the $45000$ rounds(!), whereas our algorithm outputs the same sequence of arms in every execution. In terms of the cumulative regret, our theoretical bound implies that our algorithm could be $360$ times worse compared to UCB. However, we observe that the performance overhead is of the order of $2.5$. This is encouraging news since it shows that in many instances we could get a better performance than the worst-case bound we prove. Moreover, we think that it could be possible to modify our algorithm to get a better dependence on $K$. In the current version, we pull each active arm the same number of times. However, we are trying to see how we can change that and not have a uniform number of pulls for every arm. The intuition is that if pulling a “bad” arm hurts the regret a lot, we will be able to distinguish it more easily compared to a “good” arm that does not hurt the regret by much, so pulling it more times does not lead to a big overhead in the performance. We are hopeful that this can lead to a better bound and we leave this question open for future work. We are also working on more experiments in different settings and we are planning on including them to the next version of our manuscript.
>
> >*Another related point, not necessarily a weak point but rather something arguable, is the extra dependence of $K$ in the regret bound compared to non-reproducible algorithms. It seems that the extra $K^2/\rho^2$ factor is rather difficult to be get rid of under current context. This could particularly cause problems when we have a large arm space in practice. And somehow, reproducibility makes more sense in a large scale environment. This is a question rather about the validity of the definition itself in my opinion: does it really make sense in practice? I think the authors could probably provide more discussion on that rather than (alongside some experiments ideally) stacking theoretical results for different problem settings one by one.*
>
> Our goal in this work is to provide a definition of reproducibility in interactive decision making settings that, on the one hand, provides strong reproducibility guarantees and, on the other hand, allows us to derive non-trivial regret bounds. Inspired by the work of Impagliazzo et al. (STOC’ 21), one can view our effort as a common approach in computer science where one tries to come up with a worst-case definition, see what are the optimal guarantees under this definition, and then relax it to handle beyond worst-case settings. For instance, we can define partial reproducibility with respect to a subset of the arms $K’$, meaning that the decision whether we de-activate these arms or not should be the same between the two executions. By inspecting our proof, we can see that this will lead to a $(K’)^2$ overhead instead of $K^2$. We also note that in the linear bandit setting, if the arms lie in a low-dimensional space, we have no explicit dependence on $K$ and we can handle a, potentially, infinite number of them.  Another interesting direction would be to see how we can get an instance-dependent reproducibility overhead, i.e., instead of having an extra factor that depends on $K$ to have an extra factor that depends on $\sum_{i=1}^K \Delta_i$.

---

> > ### Author Response · Authors · 2022-11-17
> > **Response to Reviewer 171s (cont.)**
> >
> > More to that, the fact that the reproducibility overhead in the regret bound does not increase with respect to $T$ is interesting, since one could imagine that as $T$ increases there are more chances that the two trajectories can differ so we would need to sacrifice performance (i.e., regret) in order to guarantee reproducibility. The main intuition for the absence of $T$ is that, as the number of rounds increases, the bandit algorithm has sufficiently explored the environment and hence is sufficiently stable in its decisions. To summarize, we hope and believe that our work serves as a non-trivial baseline for reproducibility in interactive learning settings that can inspire future research with, potentially weaker, reproducibility definitions.  Another benefit of our definition is that it makes the behavior of these algorithms more stable, interpretable, and predictable. Very often algorithms whose behavior is random make the analysis of the performance bounds tighter. For MAB, indeed, their behavior is random since it is heavily dependent on the sequence of rewards they get. However, from a practical point of view, policy makers are reluctant to use algorithms whose behavior depends heavily on randomness. We view our definition of reproducibility as a way to create decision-making algorithms whose random behavior is, in some sense, more predictable and interpretable. Notice that, for fixed internal randomness, the sequence of the pulls of the arms is distribution-dependent and not sample-dependent. In many sensitive applications like medical trials design, such properties of the underlying algorithms are crucial.
> >
> > Thank you for pointing out the typo, we will fix it.

---

> > > ### Comment · Reviewer_171s · 2022-11-20
> > > **Response to rebuttal**
> > >
> > > I pretty appreciate the author's rebuttal efforts on the experiments and providing some more intuition. To be honest, I'm ok with the paper's theoretical contribution in the first place. The main reason I gave my current score is purely because of the Program Chairs' tending to discourage ambiguous scores (e.g. by removing 4 and 7 from the scoring system and asking reviewers to discuss with ACs for 5 and 6 papers). I will update my review and score.

---

### Official Review · Reviewer_QnVR · 2022-10-25

**Confidence:** 3
**Correctness:** 4
**Technical Novelty And Significance:** 3
**Empirical Novelty And Significance:** Not applicable
**Recommendation:** 6

**Clarity, Quality, Novelty And Reproducibility:**

It certainly is an interesting problem to study from a mathematical perspective; the authors point to antecedents in the literature that underscore the importance of reproducibility. The main technical contribution (in my opinion) is showing that one can get rate-optimality of regret w.r.t. $T$ while guaranteeing reproducibility simultaneously. The paper is well written and appears comprehensive in fleshing out connections to extant literature.

Question: Is the $1/\rho^2$-scaling of the upper bounds best possible w.r.t. $\rho$? Can you please elaborate on this in the paper?

**Strength And Weaknesses:**

The paper is technically sound. It is insightful to see that while standard bandit algorithms are not reproducible in general, one can with only a slight multiplicative increase in sample complexity ensure reproducibility.

**Summary Of The Paper:**

This paper studies the stochastic multi-armed bandit problem (K-armed as well as the linear version) under "reproducibility constraints," i.e., the policy should play the same sequence of arms in any two i.i.d. instances of the problem (using the same algorithmic random seed) with probability at least $1-\rho$. The authors propose $\rho$-reproducible policies with rate-optimal regret (w.r.t. $T$).

**Summary Of The Review:**

Based on my assessment of this paper's contributions, I vote for an acceptance.

---

> ### Author Response · Authors · 2022-11-17
> **Response to Reviewer QnVR**
>
> We would like to thank the reviewer for finding our work interesting, well written, and technically sound.
>
> >*Question: Is the $1/\rho^2$-scaling of the upper bounds best possible w.r.t. $\rho$? Can you please elaborate on this in the paper?*
>
> As we mention in the conclusion, there is no reproducible algorithm that achieves instance-dependent logarithmic regret whose scaling with respect to the reproducibility parameter is $1/\rho^{2 - \varepsilon}$, for some $\varepsilon > 0$ . This follows from the lower bound that is established in Impagliazzo et al. (2021) about the number of tosses that we need to decide if a biased coin has bias $½ + c$ or $½-c$, for some small constant $c > 0$. They show that any $\rho$-reproducible algorithm needs at least $\Omega(1/\rho^2)$ samples to determine that. Let us now sketch how we can utilize this result to show our claim. We set up a bandit problem with two arms where the reward of the first arm comes from a fair coin and the reward of the second arm comes from the biased coin. Assume that there is an algorithm $\mathcal{A}$ that achieves instance-dependendent regret $O(1/\rho^{2-\varepsilon})$, for some $\varepsilon > 0.$ We run this algorithm for $T = 1/\rho^{2-\varepsilon/2}$ iterations. Notice that the algorithm will play the best arm the majority of the times, otherwise the regret will be $ \Omega(1/\rho^{2-\varepsilon/2})$, but we know that its regret is at most $O(\log(1/\rho^{2 - \varepsilon/2})/\rho^{2-\varepsilon}) < 1/\rho^{2-\varepsilon/2}.$ Since the algorithm is reproducible, this leads to a reproducible algorithm that determines the bias of the coin with less than $1/\rho^2$ flips, which is a contradiction. Hence, we can see that the $1/\rho^2$ blow-up in the regret stems from the difficulty of reproducibility deciding the bias of a coin.
>
> We will add this remark along with a more detailed explanation to the next version of our manuscript.

---

> > ### Comment · Reviewer_QnVR · 2022-11-24
> > **Post author-rebuttal**
> >
> > I thank the authors for addressing my question concisely. I will keep my score and vote for acceptance.

---

### Author Response · Authors · 2022-11-25
**General Response to Reviewers**

We would like to thank all the reviewers for taking the time to read our manuscript and our responses during the rebuttal phase. We will change our paper to incorporate all the edits they have suggested.

---

### Decision · Program_Chairs · 2023-01-20

**Decision:**

Accept: poster

**Justification For Why Not Higher Score:**

- Questions about whether this definition of reproducibility is the right one, and incomplete tie-in between the definition and how it would be used. This is compounded by an experimental section that doesn't take the opportunity to tie the theory to something concrete and practical.
- Smaller weaknesses called out in the meta-review

**Justification For Why Not Lower Score:**

- Interesting and important problem setting
- Non-trivial algorithm and analysis
- Well-written

**Metareview: Summary, Strengths And Weaknesses:**

The paper studies stochastic "reproducible bandits". A reproducible bandit algorithm is one that plays the same sequence of arms with probability at least 1-rho on any two runs of the algorithm, where the algorithm can share a random number seed across runs but the arms' rewards are generated iid.  It is easy to create a reproducible bandit --- simply pull arms in an order fully determined by the random seed --- but it isn't easy to do this while achieving low regret. Reproducibility supports debugability and interpretability. The paper studies both standard (K-armed) bandits and linear bandits.

The paper proposes rho-reproducible policies with rate-optimal regret in the time horizon. Somewhat surprisingly, while randomization seems critical in the stochastic setting, one can obtain reproducilibity with only a small multiplicative increase in regret.

Strengths
- Reproducibility is a novel and important topic
- Algorithms are interesting and clearly justified
- Regret bounds are non-trivial and technically sound
- Paper is clearly written

Weaknesses
- It is unclear about the extent to which this definition of reproducibility is truly needed to support debuggability and intepretability. Perhaps a weaker or different notion of reproducibility would be sufficient. It is also unclear whether users will value reproducibility enough to pay the regret penalty required.
- Experimental section uses only synthetic data without real-world context and is not especially compelling. While the paper is theoretical, experiments or numerical studies seem to provide an opportunity to give a more concrete support to the importance of the problem
- Regret bound has an extra dependence on K that may be problematic in large-K settings
- Novelty is somewhat reduced by the similarity of the proposed algorithms with Esfandiari et al. (2021)
- While the regret is optimal in T, there may be a gap in its dependence on other parameters


**Note From Pc:**

if the above contains the word "oral" or "spotlight" please see: "oral" presentation means -> notable-top-5% and "spotlight" means -> notable-top-25%. As stated in our emails, we are disassociating presentation type from AC recommendations